# cigChannel: A large-scale 3D seismic dataset with labeled paleochannels for advancing deep learning in seismic interpretation

Guangyu Wang[1, 2, 3], Xinming Wu[1, 2, 3], and Wen Zhang[1, 2, 3]

[1]Laboratory of Seismology and Physics of the Earth's Interior, School of Earth and Space Sciences, University of Science and Technology of China, Hefei, 230026, China
[2]State Key Laboratory of Precision Geodesy, University of Science and Technology of China, Hefei, 230026, China
[3]Mengcheng National Geophysical Observatory, University of Science and Technology of China, Mengcheng, 233500, China

**Correspondence:** Xinming Wu (xinmwu@ustc.edu.cn)

**Abstract.** Identifying paleochannels in 3D seismic volumes (seismic paleochannel interpretation) is essential for georesource development and offering insights into paleoclimate conditions. However, it remains a labor-intensive and time-consuming task. Deep learning has shown great promise in automating seismic paleochannel interpretation with high efficiency and accuracy, as demonstrated in similar image segmentation tasks in computer vision (CV). Yet, unlike the CV domain, seismic exploration lacks a comprehensive labeled dataset for paleochannels, significantly hindering the development, application, and evaluation of deep learning methods in this field. Manual labeling of paleochannels in 3D seismic volumes is tedious and subjective, potentially leading to mislabeling that degrades deep learning model's performance. To address this, we propose a workflow to generate a synthetic seismic dataset, *cigChannel*, consisting of 1,600 seismic volumes with over 10,000 labeled paleochannels. Each volume has a size of $256\times256\times256$. It is the largest dataset to date for seismic paleochannel interpretation, featuring geologically reasonable seismic volumes with accurately labeled meandering channels, tributary channel networks, and submarine canyons. A convolutional neural network (simplified from U-Net) trained on this dataset achieves F1 scores of 0.52, 0.73, and 0.63 in identifying meandering channels, tributary channel networks, and submarine canyons in three field seismic volumes, respectively. However, the synthetic seismic volumes in the *cigChannel* dataset still lack the variability and realism of field seismic data, potentially affecting the deep learning model's generalizability. To facilitate further research, we publicly release the dataset (https://doi.org/10.5281/zenodo.10791151, Wang et al., 2024), data generation codes and the trained U-Net model (https://github.com/wanggy-1/cigChannel), aiming to advance deep learning approaches for seismic paleochannel interpretation.

## 1 Introduction

Paleochannels are buried river channels that have been preserved in the geological record. They can not only provide insights into paleoclimate conditions (e.g., Leigh and Feeney, 1995; Nordfjord et al., 2005; Sylvia and Galloway, 2006), but also serve as reservoirs for groundwater (e.g., Revil et al., 2005; Samadder et al., 2011), geothermal energy (e.g., Crooijmans et al., 2016; Kang et al., 2022), ore deposits (e.g., Heim et al., 2006; Oraby et al., 2019) and hydrocarbons (e.g., Clark and Pickering, 1996; Bridge et al., 2000). Paleochannels can be identified in seismic volumes by their distinct shapes and sedimentary structures

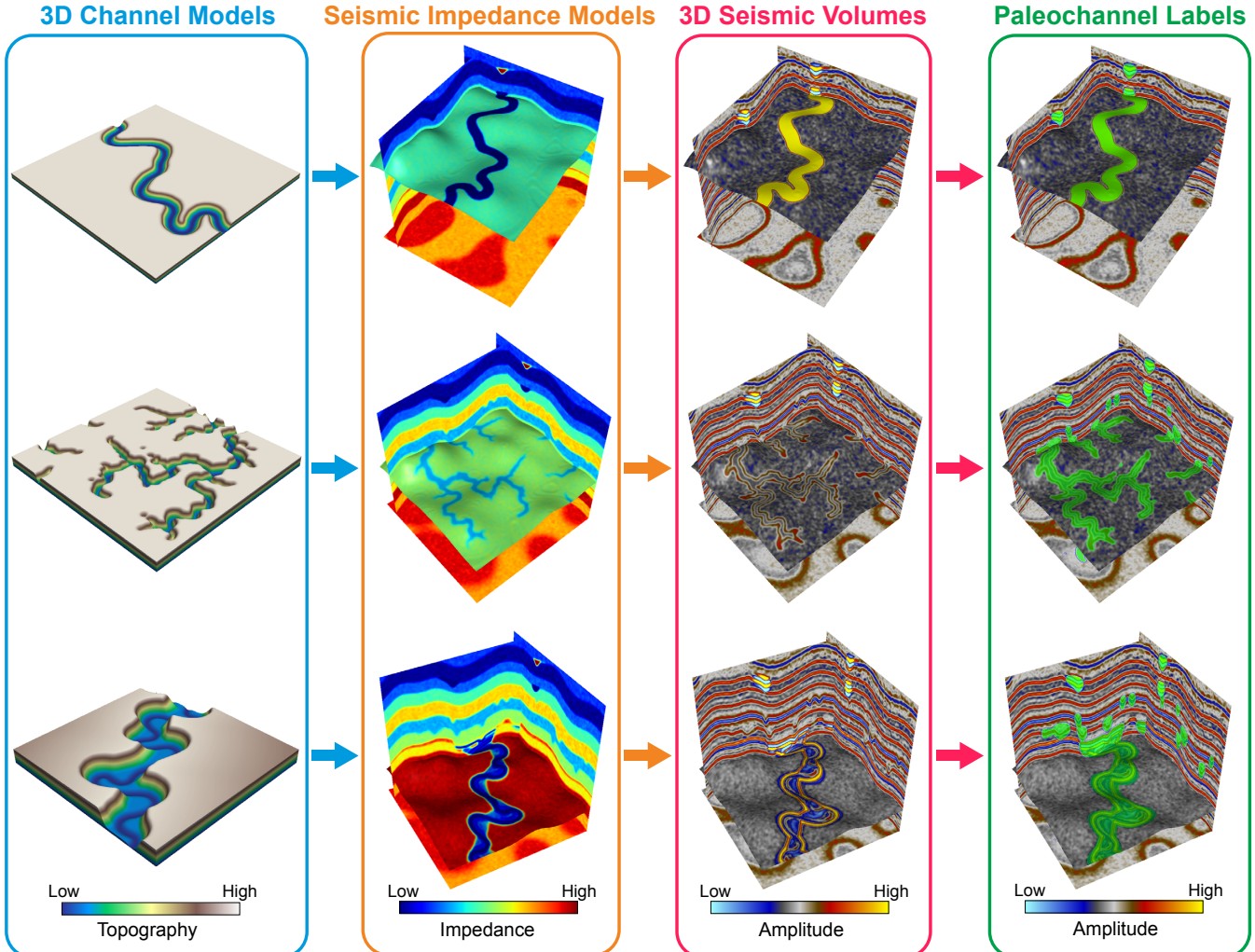

**Figure 1.** Workflow for generating the *cigChannel* dataset. First, we create 3D models of three types of paleochannels: meandering channels, tributary channel networks and submarine canyons. Second, we build 3D seismic impedance models with multiple layers and place these channels at layer boundaries as impedance anomalies. Third, the impedance models are used to calculate seismic reflection coefficients, which are subsequently convolved with Ricker wavelets to create synthetic seismic volumes. Finally, seismic reflections of paleochannels are automatically labeled. Note that the channel models, seismic impedance models and seismic volumes are all in the depth domain.

that differ from the surrounding rock formations. Although paleochannels are considered as geobodies, interpreters are limited
to view them slice-by-slice in seismic volumes. This limitation significantly increases the complexity and time of interpreting paleochannel bodies in large seismic volumes. Moreover, the historical tectonic movement may introduce deformations such as foldings to the paleochannels, making them even more difficult to recognize.

To address those issues, automatic paleochannel identification methods based on 3D convolutional neural networks (CNNs) (Pham et al., 2019; Gao et al., 2021) have been developed. The 3D CNNs are designed to capture volumetric features by performing 3D convolutions (Ji et al., 2012). They have the advantage of handling paleochannels according to their 3D nature, as opposed to the slice-by-slice visual investigation of a human interpreter. This advantage is particularly significant when the paleochannels have been deformed by historical tectonic movements (e.g., folding and faulting), which disrupt their continuity and make them more challenging to track in a slice-wise view. Another notable advantage is their efficiency. Once trained, the network can rapidly identify paleochannels in a large seismic volume. However, the main limitation of applying CNNs for paleochannel identification is the lack of labeled paleochannel samples for training. Unlike deep learning for computer vision, which benefits from numerous large datasets with labeled images such as ImageNet (Deng et al., 2009), COCO (Lin et al., 2014) and ADE20K (Zhou et al., 2017), there is no publicly available dataset of field seismic volumes with labeled paleochannels. To create such a dataset, one needs to access a large amount of field seismic volumes and correctly label the paleochannels. However, labeling paleochannels can be challenging due to the complexity of field seismic volumes, and human bias may introduce uncertainty to the labels (Bond et al., 2007). The label noise produced by mislabeling will deteriorate the performance of supervised learning (Pechenizkiy et al., 2006; Nettleton et al., 2010). Additionally, the labeling process will be time-consuming and labor-intensive.

While creating a dataset by labeling paleochannels in field seismic volumes is expensive, an alternative solution is to use synthetic seismic volumes, which are generated through a series of simulation processes in order to mimic field seismic volumes. Although lacking in sophisticated features, the synthetic seismic volumes are controllable, allowing us to tailor the features that our network will learn to segment. Moreover, mislabeling can be avoided in synthetic seismic volumes since the locations of objectives are known during the simulation process. Synthetic seismic volumes have been proven effective as training data for CNNs to identify various objectives in field seismic volumes, such as faults (Wu et al., 2019; Zheng et al., 2019), seismic horizons (Bi et al., 2021; Vizeu et al., 2022), paleokarsts (Wu et al., 2020b; Zhang et al., 2024) and paleochannels (Pham et al., 2019; Gao et al., 2021). As for paleochannel identification, the synthetic seismic datasets created by Pham et al. (2019) and Gao et al. (2021) only simulate meandering channels, while the frequently observed tributary channel networks (e.g., Nordfjord et al., 2005; García et al., 2006; Darmadi et al., 2007) and submarine canyons (e.g. Deptuck et al., 2007; Gee et al., 2007; Covault et al., 2021) are not included. Considering the diversity of paleochannels in field seismic volumes, creating a dataset with various types of paleochannels is necessary for enhancing the CNN's generalizability.

In this paper, we propose a workflow (Figure 1) for generating synthetic seismic volumes with three types of paleochannels and their labels. We first build numerous 3D models of meandering channels, tributary channel networks and submarine canyons. Parameters that control the modeling process are randomized within reasonable ranges in order to increase the diversity of channel models. Second, we build seismic impedance models with multiple layers and place the channels at layer boundaries as impedance anomalies. Third, the impedance models are used to calculate seismic reflection coefficients, which are subsequently convolved with Ricker wavelets to create synthetic seismic volumes. Finally, channels in the seismic volume can be automatically labeled since their positions are already known. Using this workflow, we have created a dataset named *cigChannel* (https://doi.org/10.5281/zenodo.10791151, Wang et al., 2024) for deep learning-based seismic paleochannel

interpretation. This dataset contains 1,600 seismic volumes and labels of more than 10,000 paleochannels. Each seismic volume has a size of 256×256×256. The effectiveness of this dataset has been validated by training a CNN to identify meandering channels, tributary channel networks and submarine canyons in three field seismic volumes, respectively. It should be noted that although we have significantly improved the diversity of paleochannels compared with previous datasets, there is no guarantee that this dataset covers every form of paleochannel in field seismic volumes. Therefore, a Python package of the dataset generation workflow (https://github.com/wanggy-1/cigChannel, see Appendix C for demonstration codes) is also provided for customizing paleochannels and facilitating further development.

## 2   Dataset generation workflow

In this section, we will outline the dataset generation workflow, covering the steps for constructing 3D channel models and synthesizing seismic volumes. We will begin by describing the modeling process for meandering channels, tributary channel networks and submarine canyons. Following that, we will explain how to build seismic impedance models based on these channel models and use the impedance models to generate synthetic seismic volumes.

### 2.1   Meandering channel modeling

Meandering channels are a common type of river channels that can be found in many seismic volumes (e.g., Noah et al., 1992; Carter, 2003; Wood, 2007; Wang et al., 2012; Alqahtani et al., 2017). They are distinguished by their sinuous paths. The continuous interaction between water and the riverbed can lead to erosion on the outer bank and deposition on the inner bank, causing the channel to migrate over time and increasing its curvature. The key to create a realistic meandering channel is to simulate its migration. We use the open-source package *meanderpy* (Sylvester, 2021) for this purpose, which employs a kinematic simulation method that computes the river migration rate as a weighted sum of upstream curvatures (Howard and Knutson, 1984; Sylvester et al., 2019). This simple kinematic model focuses on the influence of upstream curvatures on river migration and cannot capture complex processes such as compound meander formation without cutoffs (Frascati and Lanzoni, 2009). However, it is sufficient for generating morphologically realistic meandering channels. The meandering channel simulation process is demonstrated in Figure 2. We start with a straight channel with some minor perturbations, which provide initial curvatures for channel migration (Figure 2a). The channel migrates over time and forms meanders at its upstream (Figure 2b). As the migration continues, curvature of the meander increases and eventually leads to channel intersection (Figure 2c), where the channel cutoff will occur, resulting in an abandoned meander (Figure 2d). The channel migration ends when it reaches the maximum number of iteration. We neglect the abandoned channel and extract the centerline from a random segment of the most recent meandering channel, which has to be long enough to span a 256×256 square grid with a cell size of 25 m after arbitrary rotation.

The centerline is then randomly placed on the grid and rotated by a random angle between 0°and 360°. Since meandering channels in field seismic volumes typically exhibit U- or V-shaped cross-sections (e.g., Zhuo et al., 2015; Alqahtani et al., 2017; Zeng et al., 2020; Manshor and Amir Hassan, 2023), we use simplified U- or V-shaped profiles to define the channel

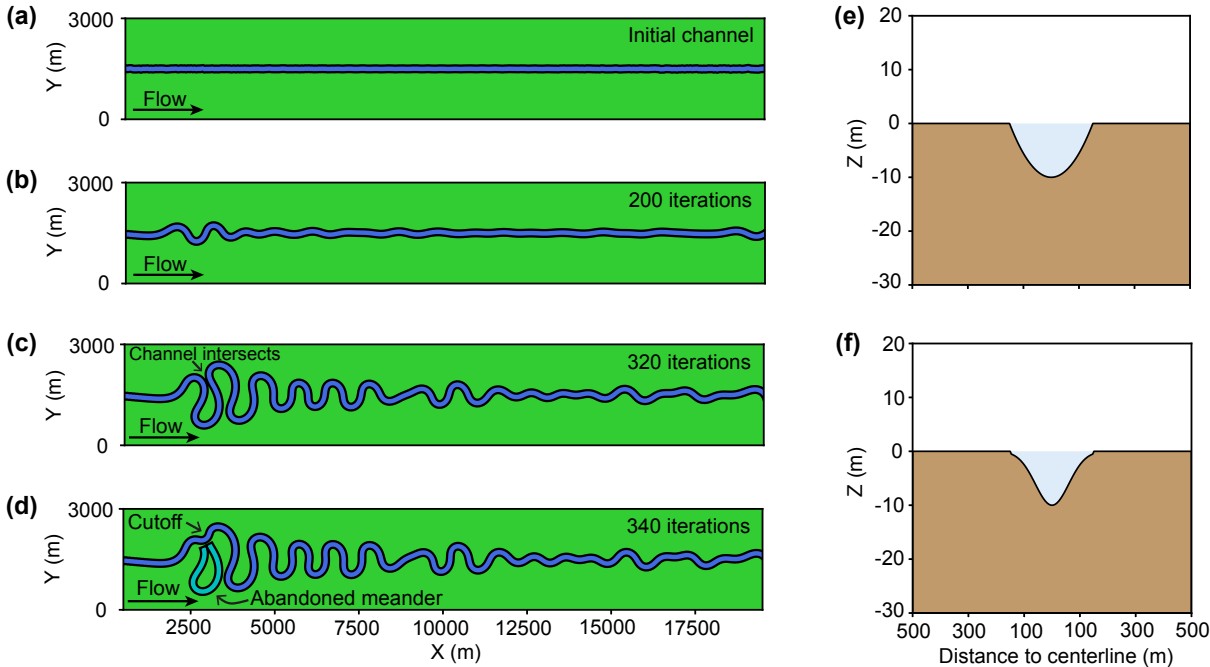

**Figure 2.** Meandering channel modeling process based on the open-source Python package *meanderpy* (Sylvester, 2021). First, we create (a) a straight channel with some minor perturbations. Then, (b) the channel begins to migrate, leading to the formation of multiple meanders. (c) The channel curvature increases as the migration continues, eventually causing a channel intersection, where (d) the channel cutoff will occur, resulting in an abandoned meander. Lastly, (e) the U- and (f) V-shaped channel cross-sections are used to define the channel topography.

topography, as shown in Figures 2e and 2f. The simplified U-shaped channel is defined as a parabolic function:

$$Z(x) = \begin{cases} 4D_c(x/W_c)^2 - D_c, & x \le W_c \\ 0, & x > W_c \end{cases}, \tag{1}$$

where $x$ is the Euclidean distance from the centerline to any point on the grid, $D_c$ is the maximum depth of the channel (which will be denoted as channel depth hereafter for simplicity) and $W_c$ is the channel width. The simplified V-shaped channel is defined as a combination of Gaussian and parabolic functions:

$$Z(x) = \begin{cases} \min[p(x), g(x)], & x \le W_c \\ 0, & x > W_c \end{cases}, \tag{2}$$

where $p(x)$ is the parabolic function in Equation 1 and

$$g(x) = -D_c e^{-\frac{x^2}{2(W_c/4)^2}}. \tag{3}$$

Although these simplified channel cross-sections may not precisely represent the real ones, they can capture their main features at a low computational cost. We create diverse topographic models of the meandering channel by randomizing the

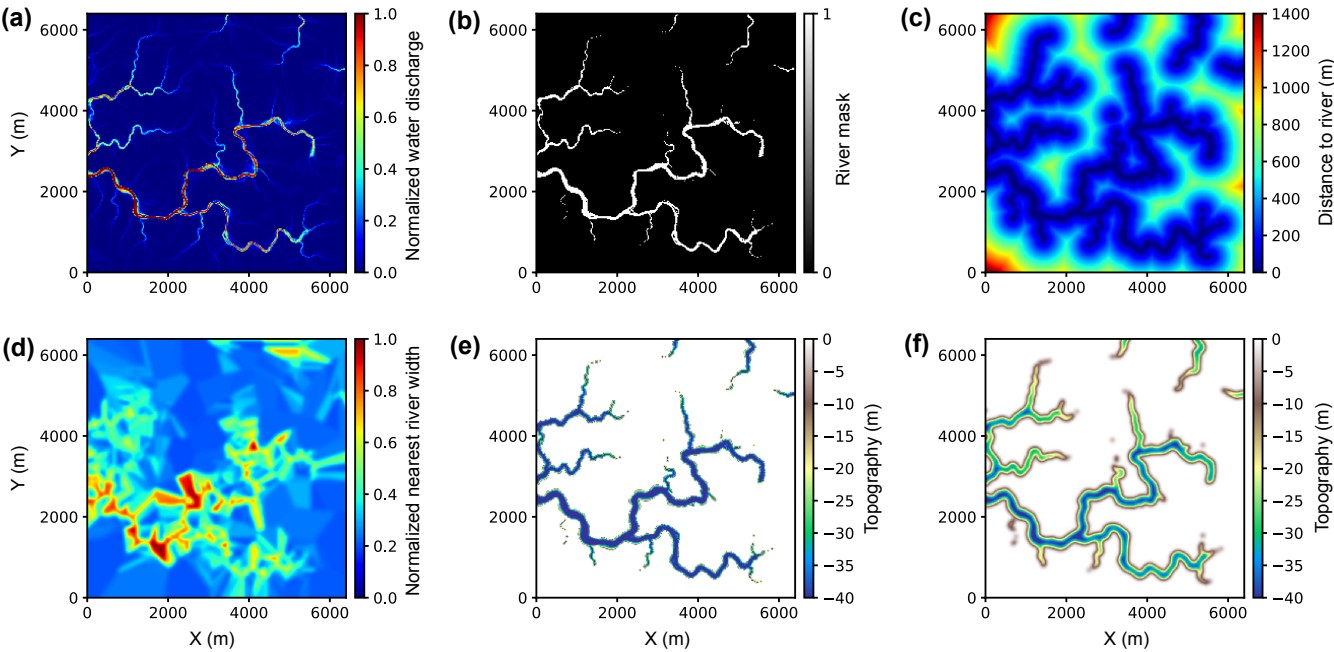

**Figure 3.** Tributary channel network modeling process based on the open-source package *soillib* (McDonald, 2020b). First, we generate (a) a map of normalized water discharge value using the *soillib* package. Second, we create (b) the river mask by binarizing the normalized water discharge with a threshold of 0.4, where discharge values greater than this threshold are considered as rivers. Third, we compute (c) the Euclidean distance to rivers and (d) the normalized nearest river width, which are subsequently used to define (e) the channel topography. Finally, to avoid abrupt topographic changes, a Gaussian filter is applied to create (f) a smoothed channel topography.

modeling parameters within reasonable ranges (see Table A1). Some examples are demonstrated in Figure 5a, showing various meandering channels with different widths, depths and meander wavelengths.

     It should be noted that in this study, we focus on identifying the most recent meandering channels in their migration histories. Therefore, all the meandering channel models only include the last channel form of the migration process. The corresponding sedimentary facies formed during the channel migration process, such as point bars, natural levees and abandoned channels (or

oxbow lakes), are not included. It is also worth noting that the width and maximum depth of each channel are fixed, while in nature they generally exhibit certain degree of variability.

### 2.2    Tributary channel network modeling

     A tributary channel network is a result of smaller rivers (tributaries) flowing into a large main river. It generally exhibits a branching or tree-like structure. To efficiently generate extensive tributary channel networks that are morphologically reasonable,

we adopt the open-source package *soillib* (McDonald, 2020b), which offers a fast implementation of particle-based hydraulic erosion that can create a morphologically reasonable tributary river network in about 10 to 20 seconds (McDonald, 2020a).

The *soillib* package is programmed to spawn hundreds of thousands of water particles at random positions on a mountainous terrain generated by random Perlin noise. The water particles move across the terrain following classical mechanics and engage in mass transfer with the surface, eventually forming a tributary river network. Figure 3a shows a map of the normalized water discharge generated by *soillib* on a 256×256 square grid with a cell size of 25m. To define the river channel topography, we first binarize the normalized water discharge by a threshold of 0.4, where discharge values greater than this threshold are considered as rivers (Figure 3b). Next, we compute the Euclidean distance from the river to each point on the grid (Figure 3c) and the normalized width of the nearest river (Figure 3d). The normalized width of the nearest river is determined by the value of normalized water discharge. Rivers with higher discharge will have larger width. We then define the channel topography using a parabolic function similar to that in Equation 1:

$$Z_i(x_i) = \min[4D_c(\frac{x_i}{W_c\alpha_i})^2 - D_c, 0], \tag{4}$$

where the subscript $i$ denotes the $i$-th point on the grid, $x$ is the distance to river, $D_c$ is the maximum channel depth, $W_c$ is the maximum channel width and $\alpha$ is the normalized width of the nearest river. The main difference between Equation 1 and 3 is the constant channel width $W_c$ in Equation 1 is replaced by a point-wise channel width $W_c\alpha_i$. By doing so, we are able to create channels with varying widths, as demonstrated in Figure 3e. The variation in channel width is controlled by $\alpha$, where the mainstream is wider and the tributaries are narrower. However, the channel topography demonstrated in Figure 3e exhibits abrupt changes at channel boundaries due to the inherent width of the river mask. Therefore, we subsequently apply a Gaussian filter to smooth the channel topography and the final result is shown in Figure 3f. When implementing the particle-based hydraulic erosion, randomness in the initial terrain and positions of water particles ensure the diversity of tributary channel networks, as demonstrated in Figure 5b. Diversity of the channel topographic models can be further increased by randomizing the maximum channel width and depth within reasonable ranges (see Table A1). Similar to the meandering channel models, the tributary channel network models are also designed for training the deep learning models to identify the final form of a tributary channel network. Therefore, these tributary channel network models do not include any sedimentary processes. As a result, our workflow only generates morphologically reasonable meandering channels and tributary channel networks. They lack stratigraphic components compared to those generated by the stratigraphic modeling methods (e.g., Flumy (Cojan et al., 2005) and Sedsim (Tetzlaff and Harbaugh, 1989)), which are more geologically realistic.

## 2.3  Submarine canyon modeling

Submarine canyons are steep-sided valleys cut into the continental shelf at the shelf/slope break (Normark et al., 1993). They are similar to river canyons on land but are formed by the movement of turbidity currents. The pathway of the turbidity current is referred to as a submarine channel. In this work, we aim at modeling a specific type of submarine canyon related to the submarine channel-levee system (Deptuck et al., 2003; Kane et al., 2007; Catterall et al., 2010), assuming the turbidity current carries enough fine-grained sediments to form natural levees. Similar to a terrestrial river channel which can meander across the floodplain on land, a submarine channel can also migrate laterally on the seabed. However, a key distinction between terrestrial and submarine channels lies in the pronounced vertical incision and aggradation of submarine channels, which

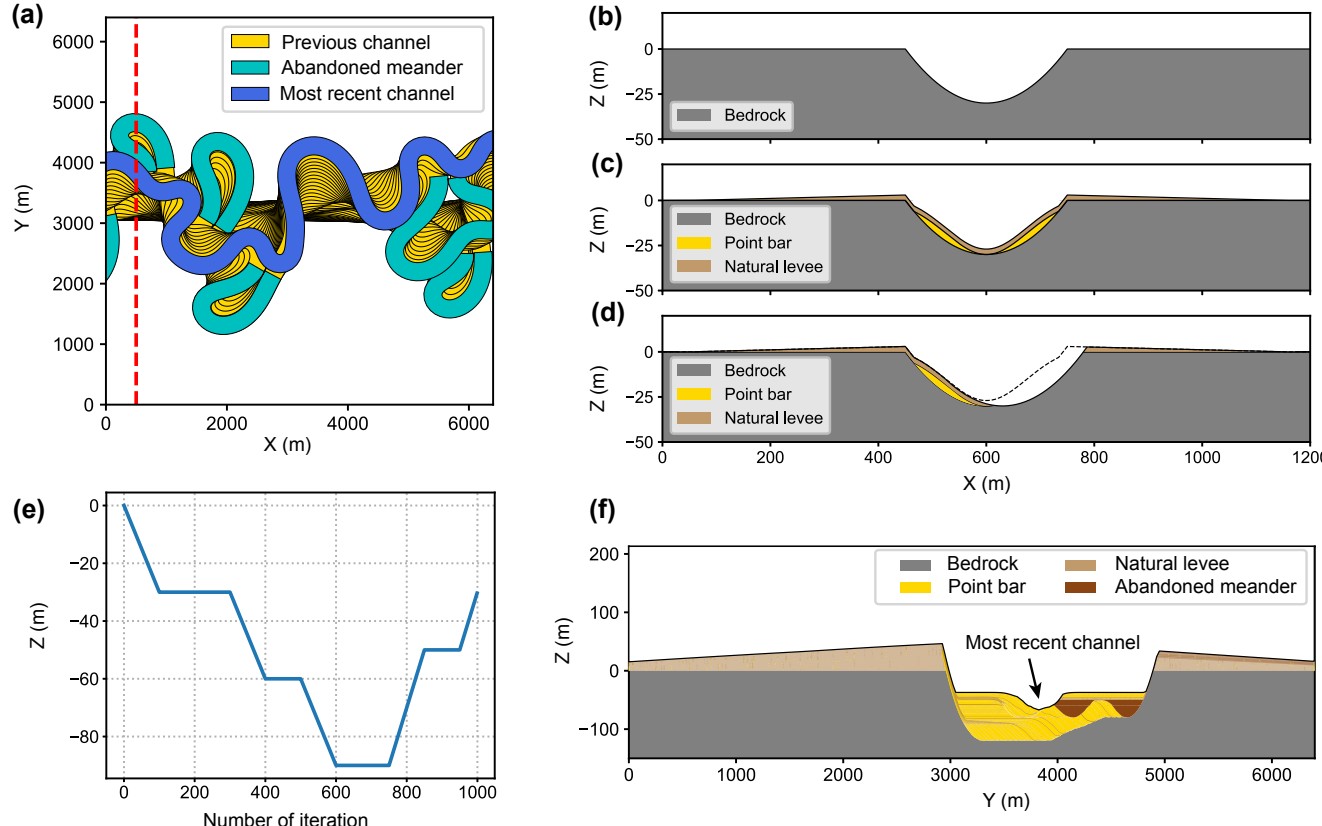

**Figure 4.** Submarine canyon modeling based on the open-source package *meanderpy* (Sylvester, 2021). (a) Lateral migration of a submarine channel. (b) Channel erosion and (c) deposition of point bars and natural levees. (d) The channel migrates towards the outer bend and erodes parts of the sediments. (e) Vertical component of the channel trajectory during the migration process, which is modified from Sylvester et al. (2011), showing an initial channel incision and a later aggradation. (f) The cross-section of the submarine canyon at the red dashed line in (a), showing a prominent erosional surface, the internal sediments and wedge-like natural levees after 1,000 iterations of channel migration.

are driven by the erosion and deposition processes associated with the turbidity current. As a result, submarine canyons are typically characterized by a prominent erosional surface and internal layered sediments, which can be clearly observed in high-resolution seismic profiles (Kolla et al., 2007).

To model the erosional surface and internal layered sediments of a submarine canyon, we adopt a modeling method based on submarine channel trajectories (Sylvester et al., 2011), which is also implemented in *meanderpy*. The modeling process

is illustrated in Figure 4. It first simulates the lateral migration of a submarine channel (Figure 4a). At each iteration during the migration process, a parabolic function (Equation 1) is used to define the surface of channel erosion (Figure 4b), which is followed by deposition of point bars and natural levees (Figure 4c). Point bars are accumulated sediments on the inner bends of the channel where the flow velocity is lower. Their top surface is defined using a combination of parabolic and Gaussian

function (Equation 2 and 3). For the convenience of modeling, point bars are created on both inner and outer bends, with those on the outer bends will be subsequently eroded. Natural levees are structures that form along the sides of a submarine channel when the turbidity current overflow the channel banks. They typically exhibit a wedge-like shape because the turbidity current loses energy and sediments as it move away from the channel margins. The thickness of the natural levee is defined as follows:

$$
T(x) = \begin{cases} \frac{T_{\max}}{W_l}\left(x - \frac{W_c - W_l}{2}\right), & x \geq W_c \\ T_{\max}, & x < W_c \end{cases},
\tag{5}
$$

where $x$ denotes the distance to channel centerline, $T_{\max}$ is the maximum thickness of the levee, $W_l$ is the levee width and $W_c$ is the channel width. After the deposition of point bars and natural levees, the channel will migrate towards its outer bends and erode parts of these sediments (Figure 4d). The erosion and deposition processes repeat until the channel migration ends. In the meantime of lateral migration, the channel also experiences vertical incision and aggradation (Figure 4e). When the channel migration ends, a submarine canyon will be created, along with wedge-like outer levees and internal layered sediments. The internal sediments consist of interbedded layers of point bars and inner levees, as well as abandoned meanders (Figure 4f). To create diverse forms of submarine canyons, we use a random set of modeling parameters within reasonable ranges (see Table A1). Some of the submarine canyon models are demonstrated in Figure 5c.

## 2.4 Seismic volume simulation

After constructing over 10,000 channel topographic models covering meandering channels, tributary channel networks and submarine canyons, we proceed to create synthetic seismic volumes based on these models. The first step is to define the seismic impedance, which is a crucial parameter for simulating seismic events. In seismic exploration, seismic waves from an artificial source travel through the subsurface rock mass, and part of the waves will be reflected back to the surface at the boundaries of two geologic layers with a contrast in seismic impedance. The reflected seismic waves will form seismic events, which reflect the geometry of layer boundaries. The amplitudes of the seismic events are related to the contrast in seismic impedance between two layers. We start by generating 3D seismic impedance models with horizontal layers. In each layer, we add some minor random perturbations to the impedance to make it more realistic. Details about the configuration of the impedance model are listed in Table D1. The channel topographic models are then placed at the layer boundaries, and the seismic impedance of the channel is defined according to the channel type.

For meandering and tributary channels, we fill them with a relatively uniform impedance with some minor perturbations (approximately 100 m/s.g/cm$^3$). The average impedance of the channel is determined by a parameter $\varepsilon$, which is defined as the impedance contrast between the channel and its overlying layer:

$$
\varepsilon = \frac{|Z_f - Z_u|}{Z_u},
\tag{6}
$$

where $Z_f$ and $Z_u$ denotes the impedance of the channel and its overlying layer, respectively. The value of $\varepsilon$ varies between zero and one. $\varepsilon = 1$ indicates the highest impedance contrast between the channel and its overlying layer, and $\varepsilon = 0$ indicates

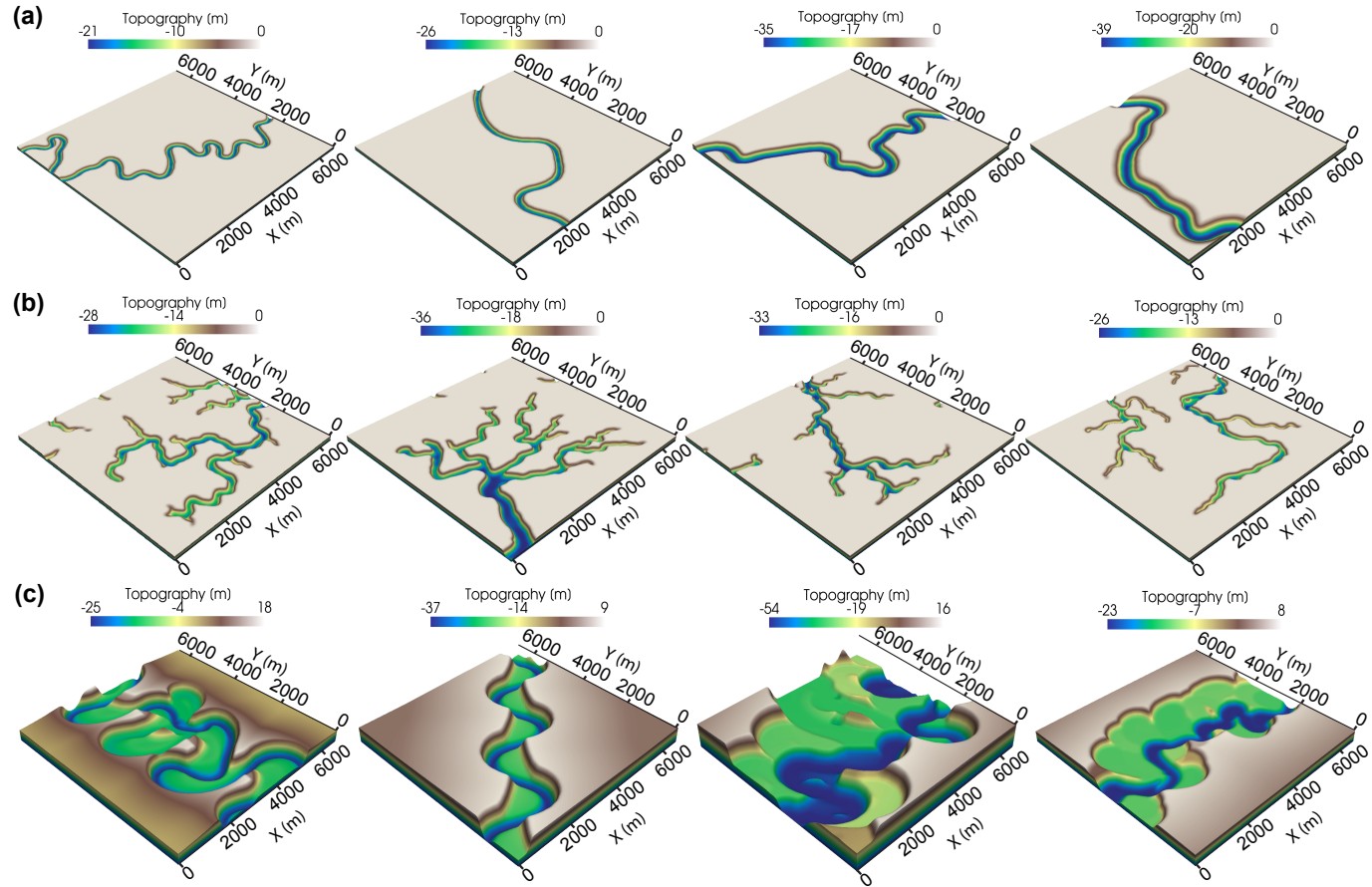

**Figure 5.** Diverse topographic models of (a) meandering channels, (b) tributary channel networks and (c) submarine canyons.

equal impedance between the channel and its overlying layer. Figures 6a and 6b demonstrate the horizontal and vertical slices of a 3D impedance model, which consists of meandering and tributary channels with a high impedance contrast ($\varepsilon = 1$). The impedance model is then used to calculate seismic reflectivity using the following equation:

$$R_i = \frac{Z_{i+1} - Z_i}{Z_{i+1} + Z_i}, i = 1, 2, ..., N - 1, \tag{7}$$

where the subscript $i$ denotes the $i$-th point in the vertical (depth) direction of the model, and $N$ denotes the total number of points in the vertical direction. The reflectivity model is then convolved with a Ricker wavelet (see Figures 8a and 8b for examples), which is commonly used to create synthetic seismic data. The mathematical expression of a Ricker wavelet in the depth-domain is:

$$f(s) = (1 - 2\pi^2 k_m^2 s^2)e^{-\pi^2 k_m^2 s^2}, \tag{8}$$

where $s$ denotes the distance and $k_m$ denotes the peak wavenumber of the wavelet. Figure 6c shows the seismic volume with a high impedance contrast between the channel and its overlying layer. The channels exhibit strong seismic amplitudes as shown

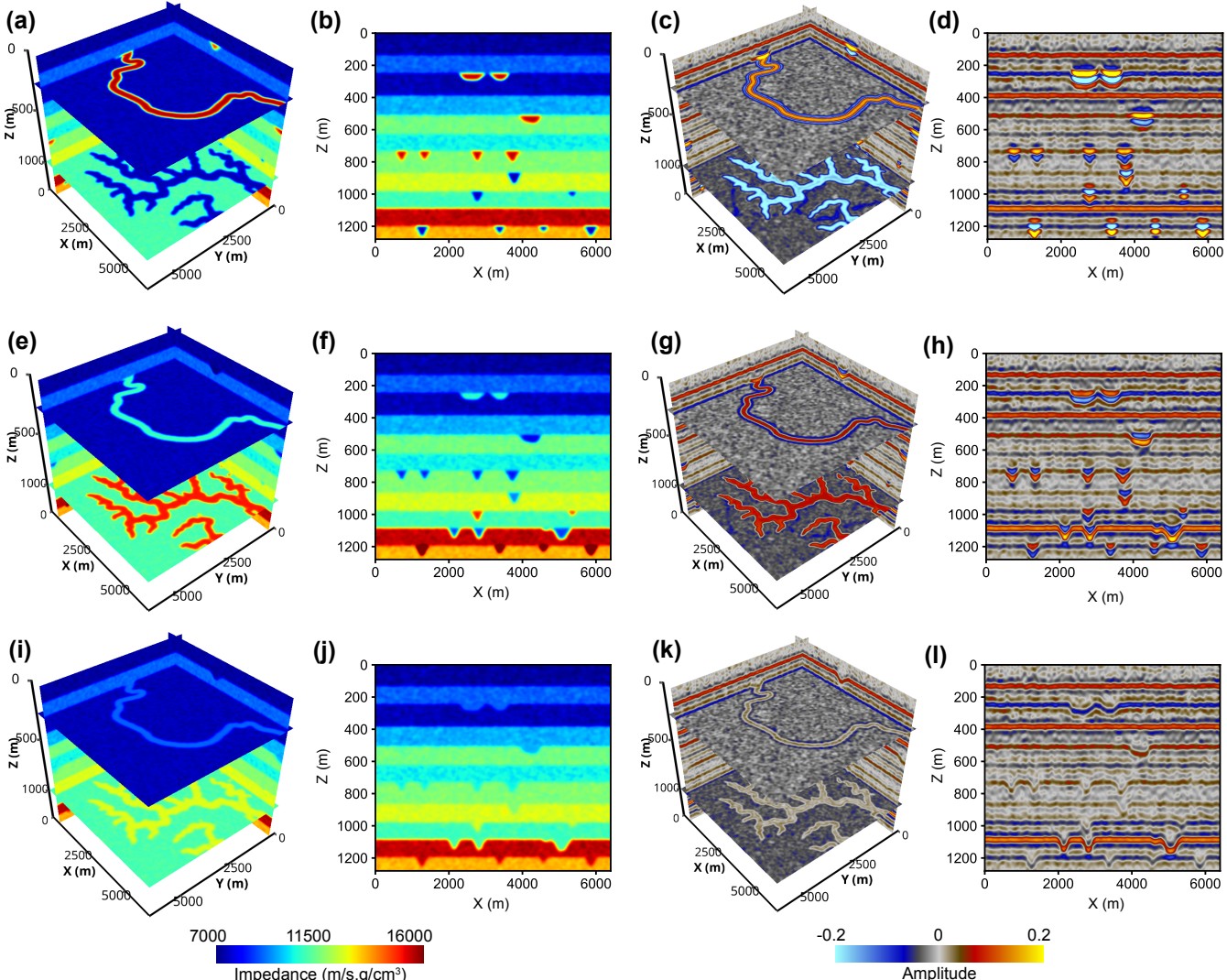

**Figure 6.** Seismic impedance and amplitude volumes containing meandering channels and tributary channel networks, showing different levels of impedance contrast between channels and their overlying layers. (a) to (d) correspond to channels with a high impedance contrast, (f) to (h) correspond to channels with a low impedance contrast, and (i) to (l) correspond to channels with no impedance contrast with their covering layers.

in Figure 6d. As $\varepsilon$ decreases to 0.2, the impedance contrast between the channel and its overlying layer becomes lower, as shown in Figures 6e and 6f. The channel shows reduced seismic amplitude (Figure 6g), appearing as infilling characteristics on the seismic profile (Figure 6h). When $\varepsilon = 0$, the impedance of the channel will be the same as that of its overlying layer (Figures 6i and 6j). As a result, the channels show no internal seismic reflections except along their boundaries (Figure 6k), forming U- or V-shaped reflection patterns on the seismic profile (Figure 6l).

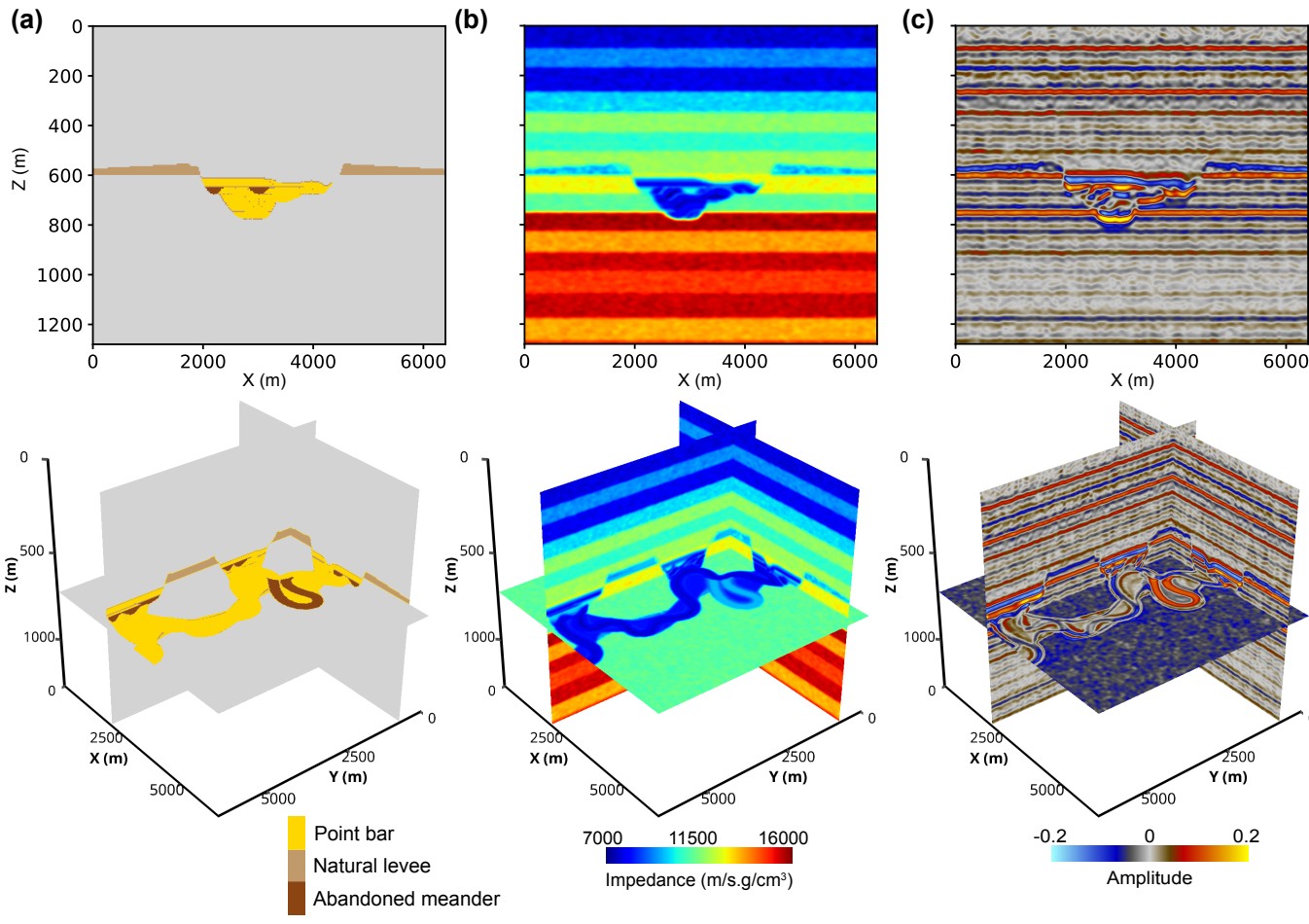

**Figure 7.** (a) Sedimentary facies, (b) seismic impedance and (c) seismic amplitude of a submarine canyon.

The impedance of a submarine canyon is determined based on its sedimentary facies. Figure 7a shows the sedimentary facies of a submarine canyon, including point bar, natural levee and abandoned meander. As shown in Figure 7b, point bars are assumed to be sand-rich and are assigned a lower seismic impedance, whereas natural levees and abandoned meanders are considered mud-rich and are assigned a higher impedance. The impedance range for each sedimentary facies is listed in Table D1. The impedance contrasts between adjacent layers within the point bars result in internal seismic reflections that appear as layered patterns on seismic profiles and meander-belt-like patterns on horizontal slices, as shown in Figure 7c. Minor impedance perturbations ($\pm 100$ m/s.g/cm$^3$) also exists within each sedimentary facies.

In the current impedance model, all the channels and layers are modeled as horizontal. However, in real-world settings, channels and layers often undergo structural deformation (e.g., folding and faulting), which are commonly observed in many field seismic volumes. To increase the diversity and realism of synthetic seismic volumes, we introduce inclination, folds and

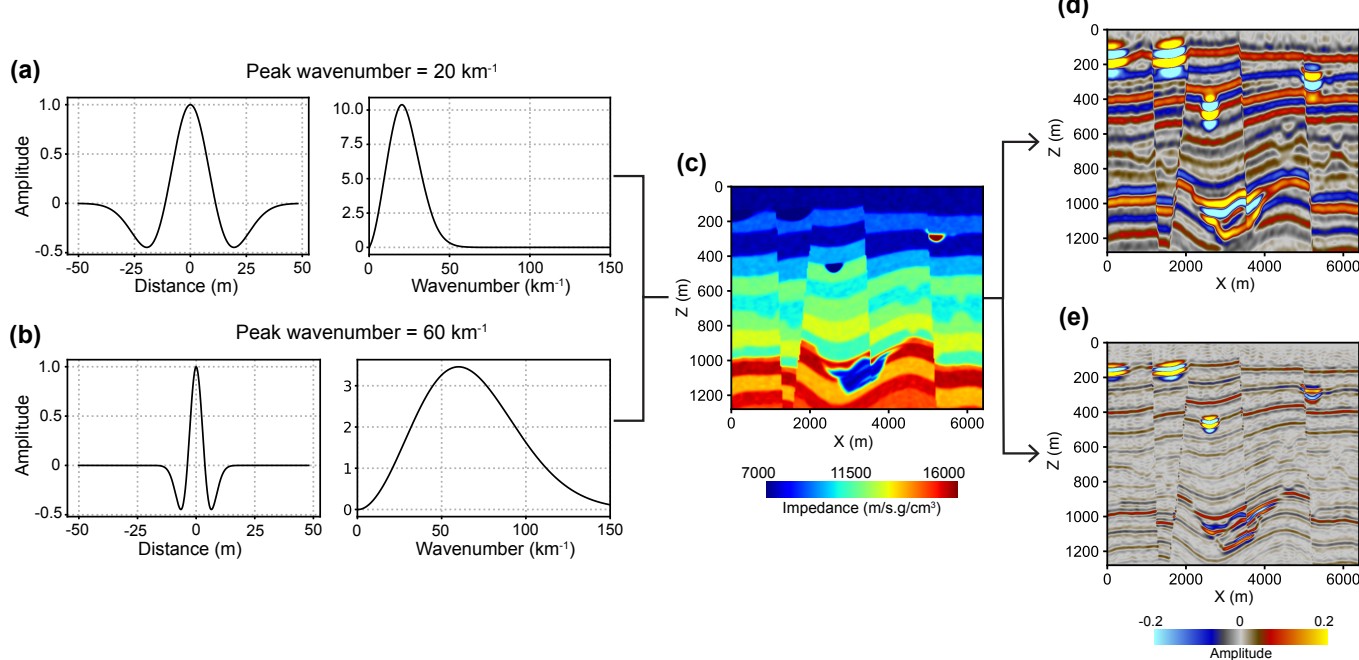

**Figure 8.** Synthetic seismic profiles with different wavelets computed from the same seismic impedance model. (a) A small-wavenumber Ricker wavelet with a peak wavenumber of 20 km$^{-1}$ (corresponding to a peak frequency of 20 Hz) in depth and wavenumber domain. (b) A large-wavenumber Ricker wavelet with a peak frequency of 60 km$^{-1}$ (corresponding to a peak frequency of 60 Hz) in depth and wavenumber domain. (c) Seismic impedance model. (d) Low-resolution seismic profile generated by using the small-wavenumber wavelet. (e) High-resolution seismic profile generated by using the large-wavenumber wavelet.

faults into the impedance model following the workflow proposed by Wu et al. (2020a). An example of the resulting impedance model with structural deformation is shown in Figure 8c. Another way to increase the diversity of the synthetic seismic volumes is to use wavelets with various peak wavenumbers. This consideration is also important, as the peak wavenumber of seismic reflections from the channel can vary significantly in field seismic volumes, depending on various factors such as

the absorption effects of subsurface media and the characteristics of the seismic source (Yilmaz, 2001). Figure 8 shows two synthetic seismic profiles generated using different wavelets but the same impedance model. A wavelet with a smaller peak wavenumber (Figure 8a) results in a low-resolution seismic profile with broad seismic events (Figure 8d), where some thin layers within the submarine canyon at the bottom of the profile are difficult to distinguish. In contrast, using a wavelet with a larger peak wavenumber (Figure 8b) produces a high-resolution seismic profile (Figure 8e), in which those thin layers within

the submarine canyon become clearly discernible. The peak wavenumber ranges of the Ricker wavelets used to generate the synthetic seismic volumes are listed in Table D1.

## 3 Results

Using the proposed workflow, we construct the *cigChannel* dataset (https://doi.org/10.5281/zenodo.10791151, Wang et al., 2024), which consists of 1,600 synthetic seismic volumes with over 10,000 labeled paleochannels. Each seismic volume has a size of 256×256×256. This dataset is organized into four subsets, corresponding to different channel types: meandering channels, tributary channel networks, submarine canyons, and assorted channels. Examples of seismic volumes and paleochannel labels in these four subsets are demonstrated in Figure 9. In addition to the seismic volumes, the *cigChannel* dataset also provides the corresponding seismic impedance models, as illustrated in Figure S1 of the Supplementary Material. Furthermore, the submarine canyon subset provides sedimentary facies volumes corresponding to the submarine canyon, with illustrative examples shown in Figure S2 of the Supplementary Material. A detail breakdown of the dataset's components is presented in Table B1.

For training deep learning models to identify specific types of channels, the subsets of meandering channels, tributary channel networks and submarine canyons each provide 400 seismic volumes containing only the corresponding type of channel. Each subset provides binary class labels, where 0 denotes non-channel areas and 1 denotes channels. As shown in Figure 9, each subset contains seismic volumes with horizontal, inclined, folded, and faulted structures, aiming to facilitate the training of deep learning models to identify channels associated with various structural styles. These structures are randomly generated to introduce variability into the seismic volumes. Since submarine canyons are generally wider and deeper than terrestrial channels (Normark et al., 2003; Kolla et al., 2007; Covault et al., 2021), this characteristic is reflected in the *cigChannel* dataset by generating submarine canyons that are larger than meandering and tributary channels.

The assorted channel subset also has 400 seismic volumes. Each seismic volume contains multiple terrestrial channels (including meandering channels and tributary channel networks) and one submarine canyon. This subset provides multi-class labels of non-channel areas, terrestrial channels and submarine canyons, as shown in Figure 9d. They are denoted by 0, 1, and 2 in the label volume, respectively. The reason of simulating multiple terrestrial channels but only one submarine canyon in a single seismic volume is to balance their voxel amounts, since a model trained on an imbalanced dataset perform poorly on the minority class (i.e., the class-imbalance problem). However, there is still a huge gap in voxel amounts between channels and non-channel areas. This gap exists in all four subsets. Therefore, we suggest to adopt strategies for addressing the class-imbalance problem when using the *cigChannel* dataset to train a deep learning model, such as employing the class-balanced cross-entropy loss function (Xie and Tu, 2015).

## 4 Applications

Three U-Nets are trained on the subsets of meandering channels, tributary channel networks and submarine canyons, respectively, which are then applied to identify paleochannels in field seismic volumes. The U-Net architecture used in this study is shown in Figure 10, where the number of convolutional layers and feature maps is reduced from the original design in Ronneberger et al. (2015) to reduce memory usage and computational cost. The network's input is a 224×224×224 seismic volume, which is cropped from the original 256×256×256 volume due to the memory limit of GPU. Each seismic volume is normalized

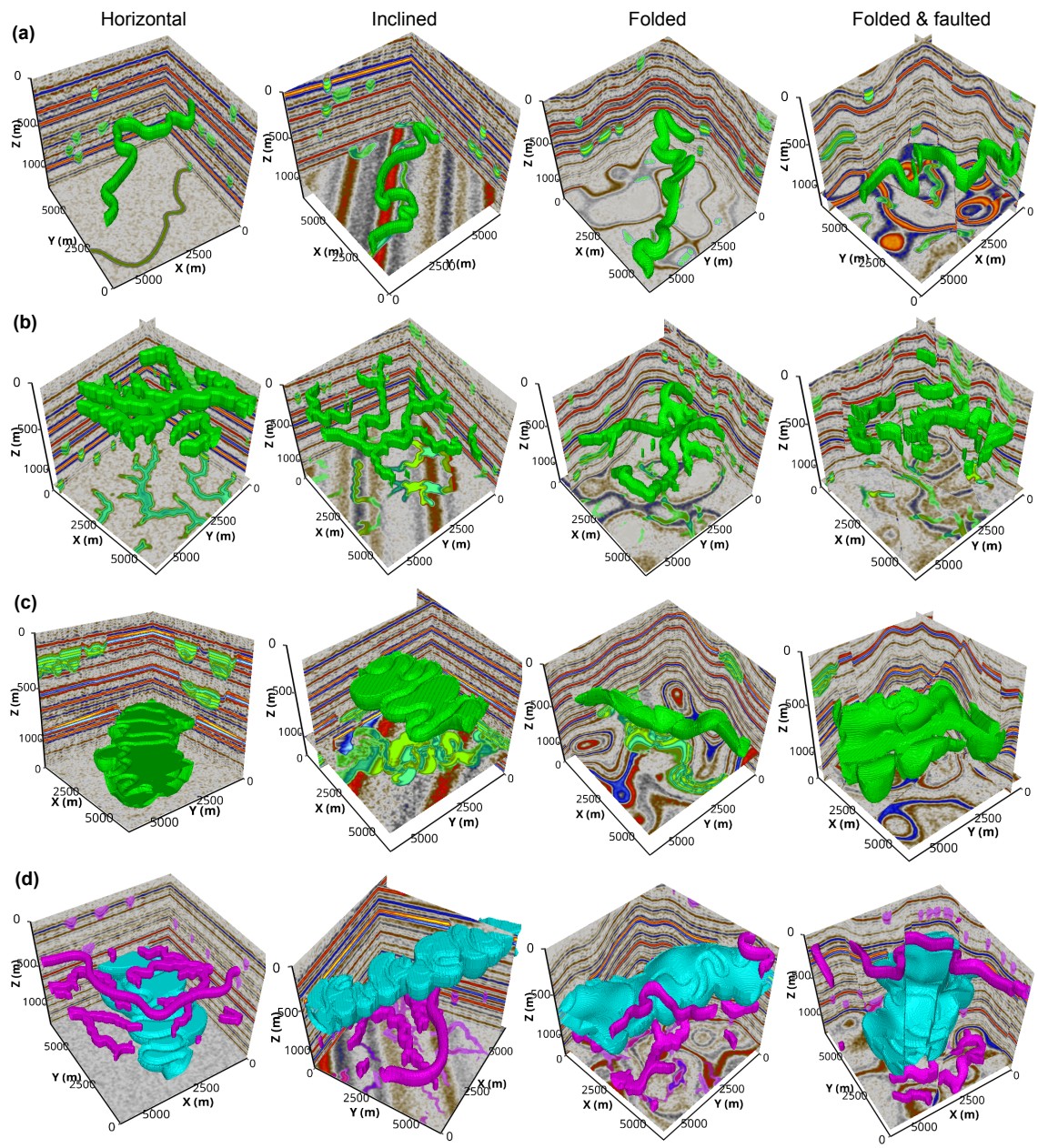

**Figure 9.** Synthetic seismic volumes and paleochannel labels in the (a) meandering channel, (b) tributary channel network, (c) submarine canyon and (d) assorted channel subsets of the *cigChannel* dataset, showing various types of structures. The first three subsets provides binary class labels to distinguish between channels and the background (i.e. the non-channel areas), while the assorted channel subset provides multi-class labels to distinguish between terrestrial channels, submarine canyons and the background.

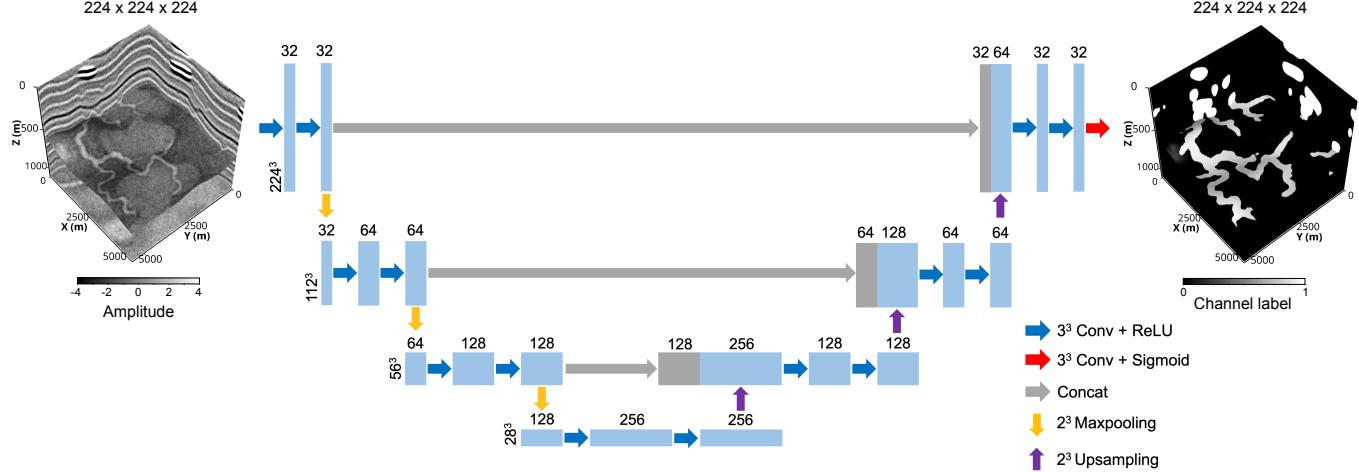

**Figure 10.** A simplified U-Net for identifying paleochannels in seismic volumes. The inputs of the U-Net are seismic volumes and the outputs are channel probabilities between 0 and 1.

using the mean-variance normalization method, and Gaussian random noise is added to the synthetic seismic volume to make the training process more robust and reduce the tendency towards overfitting. The added noise is zero-mean and its standard deviation is determined according to the expected signal-to-noise ratio (SNR) of the noisy seismic volume. We set the SNR of each seismic volume to vary between 5 dB and 10 dB, which is a reasonable range for field seismic volumes (Zhang et al., 2017; Wu et al., 2021). The noisy seismic volume goes through the contracting and expansive path of the U-Net for feature extraction. The final output layer of the network is a $3\times3\times3$ convolutional layer followed by a sigmoid activation, which maps the extracted feature into channel probabilities between 0 and 1. We binarize the channel probabilities using a threshold of 0.5 in order to obtain a binary segmentation of channels and non-channel areas.

To evaluate the training performance, each subset is divided into training and test sets. The training and test sets contain 70% and 30% of the total samples, respectively. The class-balanced cross-entropy is used as loss function regarding the huge gap in voxel amounts between channels and non-channel areas. The F1 score is used as a metric to evaluate the network's performance on the test set. We use the Adam method (Kingma, 2014) to optimize the network's parameters and set the learning rate to be 0.0001. As shown in Figure 11, the training loss of each network converges after 200 epochs, and the F1 scores of the test sets gradually increase to around 0.9. The networks from the final epoch are used to identify paleochannels in field seismic volumes.

The first U-Net is trained on the meandering channel subset and applied to a volume from the Parihaka seismic survey (https://wiki.seg.org/wiki/Parihaka-3D). As shown in Figure 12a, the seismic volume reveals several meandering channels feeding into a larger channel (may be a submarine canyon). The channel identification result of the U-Net is shown in Figure 12b, which has a F1 score of 0.52 when compared to the human-made channel interpretation (Figure 12c). Some channel areas

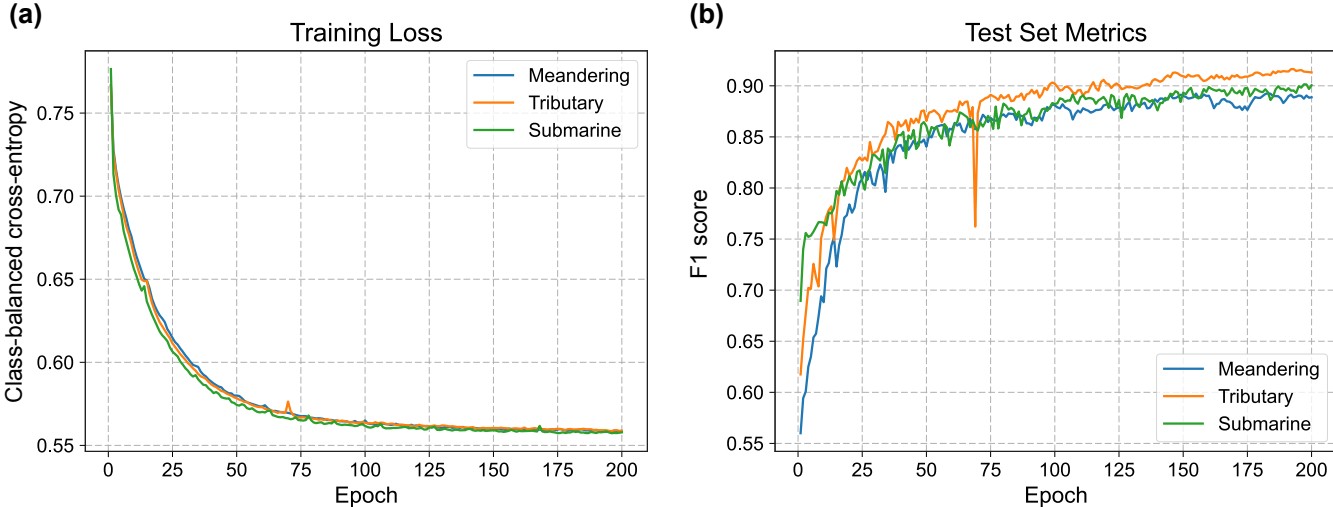

**Figure 11.** Training progress of the U-Net on the subsets of meandering channels, tributary channel networks and submarine canyons, showing the (a) training loss (class-balanced cross-entropy) and (b) F1 score on the test set over epochs.

with significant variations in seismic amplitude or where the channel width suddenly increase are not correctly identified, as indicated by the blue arrows in Figure 12b. This is likely due to that each meandering channel in the training set has a fixed channel width, and the seismic amplitude within each channel is relatively uniform. Moreover, there are many false positive channel identification results, as indicated by the green arrows in Figure 12b, which might be local structural deformations that resemble the feature of a U- or V-shaped channel.

The second U-Net is trained on the tributary channel network subset and applied to a volume from an anonymous seismic survey (denoted as NW seismic survey hereafter). As demonstrated in Figure 13a, this seismic volume shows a tributary channel network with V-shaped channel cross-sections. Seismic amplitudes within the channel are relatively homogeneous, indicating a relatively uniform seismic impedance within the channel as we designed in our data generation workflow. The channel identification result of the U-Net is demonstrated in Figure 13b, showing that most of the channels are correctly identified. However, there are still a number of small-scale structural deformations that are mistakenly identified as channels, as indicated by the green arrows in Figure 13b. The F1 score between the U-Net and human-made interpretation result (Figure 13c) is 0.73.

The last U-Net is trained on the submarine canyon subset and applied to another volume from the Parihaka seismic survey. As demonstrated in Figure 14a, a submarine canyon characterized by a prominent erosional surface is observed in the seismic volume. It has a relatively low seismic amplitude compared with that of its surrounding layers, indicating a low discrepancy in seismic impedance within the canyon. However, some layered structures are still visible within the canyon. Figure 14b demonstrates the channel identification result of the U-Net. Most areas of the submarine canyon are correctly identified but

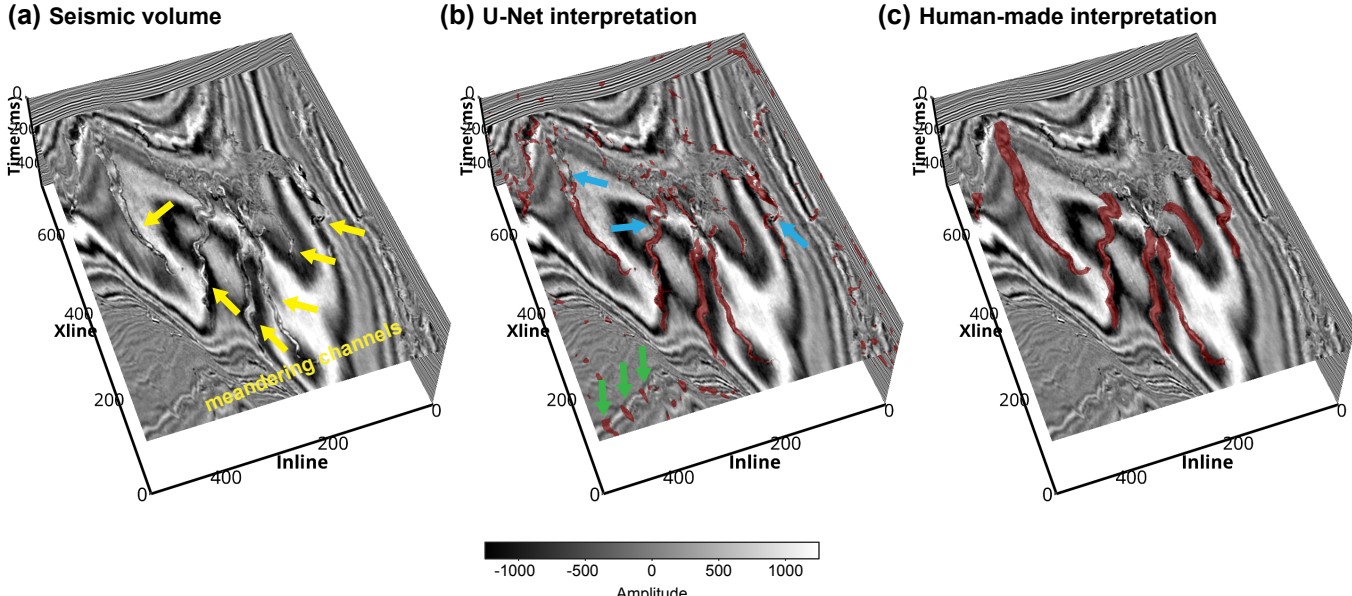

**Figure 12.** (a) Seismic volume from the Parihaka seismic survey (courtesy of New Zealand Crown Minerals), showing multiple meandering channels (indicated by the yellow arrows). (b) Channel interpretation result of the U-Net trained on the subset of meandering channels. The blue arrows indicate channel areas that fail to be identified, and the green arrows indicate false positive channel identification results. (c) Human-made channel interpretation result.

the U-Net cannot delineate the canyon boundary accurately. The F1 score between the U-Net and human-made interpretation result (Figure 14c) is 0.63.

## 5  Discussion

### 5.1  Plausibility of the synthetic seismic volumes

While the *cigChannel* dataset provides various samples for training deep learning models to identify paleochannels in seismic volumes, the plausibility of the synthetic seismic volume remains uncertain. Several simplifications are applied to reduce computational costs during the generation of synthetic seismic volumes. For instance, the configuration of seismic impedance models ignores the variability within layers and channel facies. However, this variability is ubiquitous in the subsurface. Moreover, the forward seismic modeling uses the simplest 1D convolution between seismic (P-wave) impedance and Ricker wavelet. It disregards many aspects of wave propagation in the subsurface, including the contribution of shear waves, separate contributions from P-wave velocity and density, and multi-path reflection. These simplifications reduce the realism of synthetic seismic volumes. It is questionable whether the synthetic seismic volumes can capture the patterns in the field seismic volumes.

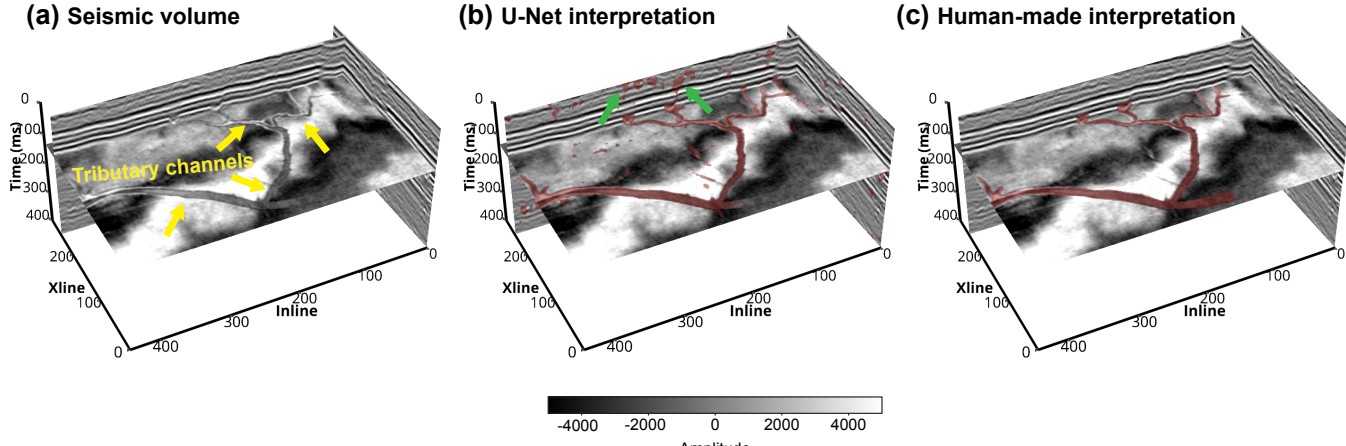

**Figure 13.** (a) Seismic volume from an anonymous seismic survey (denoted as NW seismic survey), showing a tributary channel network (indicated by the yellow arrows) with V-shaped channel cross-sections. (b) Channel interpretation result of the U-Net trained on the subset of tributary channel networks. Some false positive channel interpretation results are indicated by the green arrows. (c) Human-made channel interpretation result.

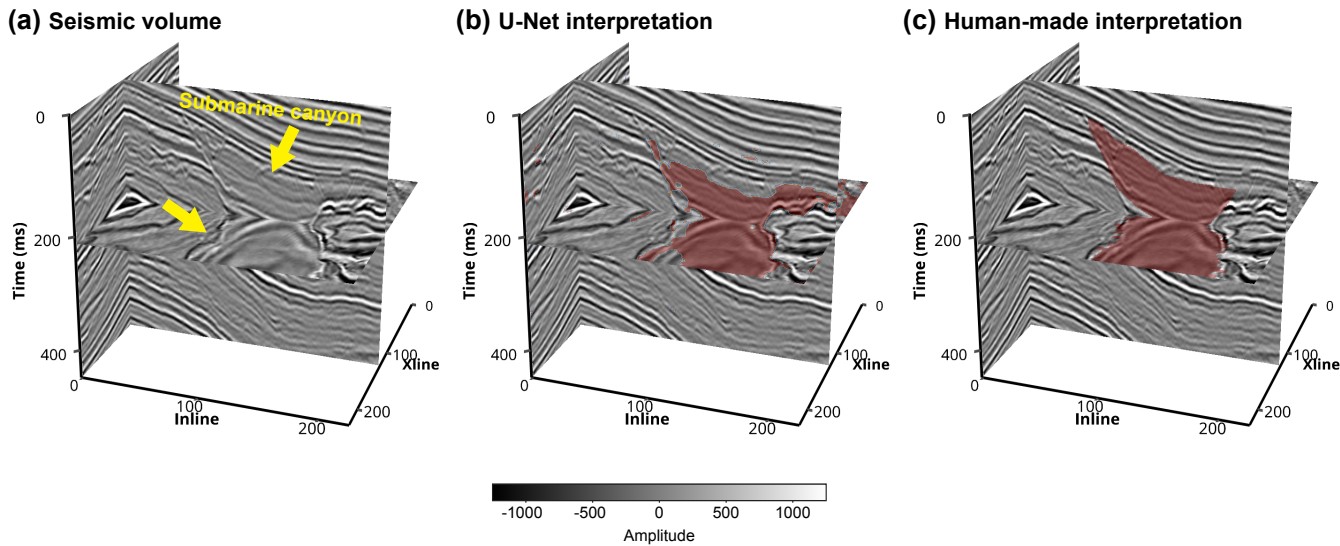

**Figure 14.** (a) A field seismic volume from the Parihaka seismic survey (courtesy of New Zealand Crown Minerals), showing a submarine canyon (indicated by the yellow arrows). (b) Channel interpretation result of the U-Net trained on the subset of submarine canyon. (c) Human-made channel interpretation result.

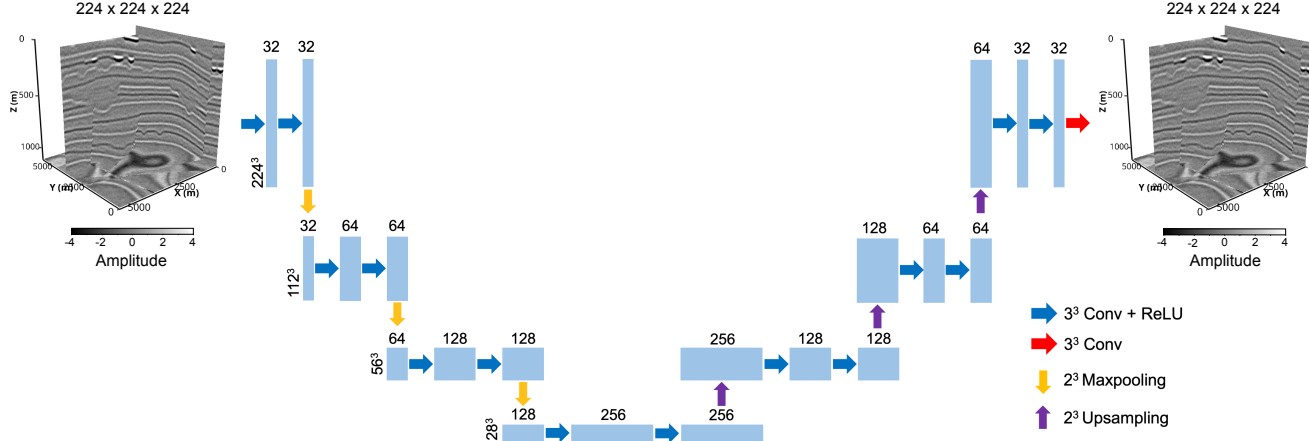

**Figure 15.** U-Net-based autoencoder architecture for reconstructing seismic volumes. Compared to the U-Net architecture used for paleochannel identification, the skip connections are removed, and the final layer is a 3×3×3 convolutional layer without sigmoid activation. The inputs of the autoencoder are the original seismic volumes and the outputs are their reconstruction results.

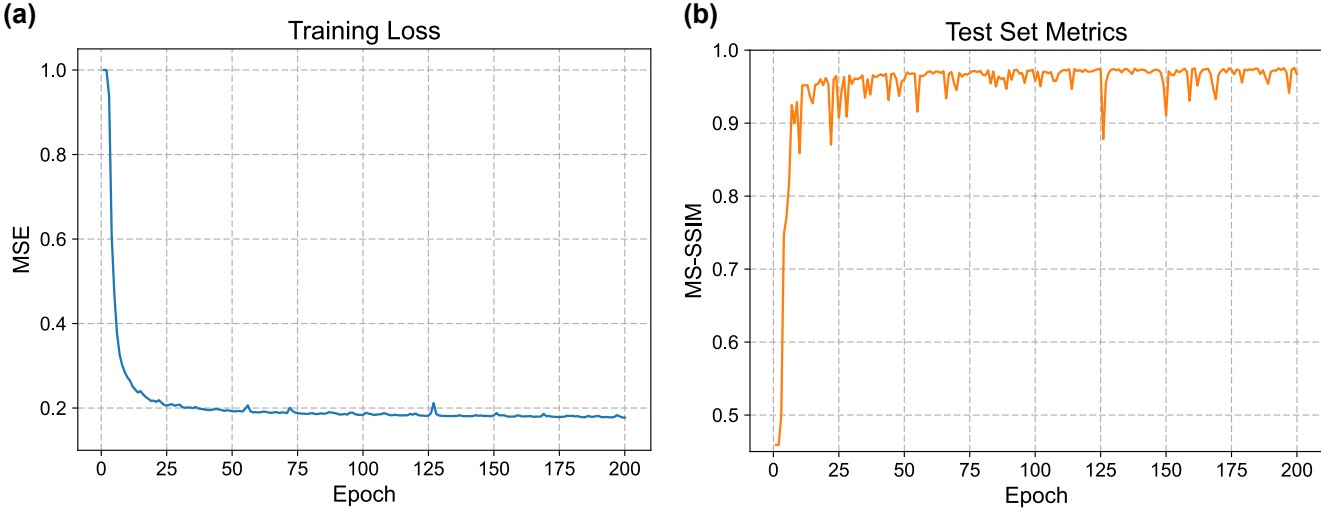

**Figure 16.** Training progress of the U-Net-based autoencoder, showing (a) training loss (mean squared error) and (b) multi-scale structural similarity (MS-SSIM) on the test set over epochs.

To answer this question quantitatively, we use the synthetic seismic volumes in the *cigChannel* dataset to train an autoencoder to reconstruct seismic volumes. If this autoencoder can reconstruct the field seismic volumes as well as the synthetic ones, it means that the synthetic seismic volumes are plausible and representative enough of field seismic volumes. Otherwise, it

indicates room for improvement. To construct training and test set, we randomly choose 70 samples for training and 30 samples for testing from each subsets. That makes a total number of 280 training samples and 120 test samples. The architecture of the autoencoder is adapted from the U-Net used for identifying paleochannels. As shown in Figure 15, we remove all the skip connections from the U-Net and the sigmoid activation from the final convolutional layer. Each synthetic seismic volume is cropped into a size of 224×224×224. They will serve as both inputs and labels to train the autoencoder. The seismic volumes (both synthetic and field ones) will be normalized and zero-mean Gaussian random noise will be added to the synthetic seismic volume. The standard deviation of the noise is determined according to the expected SNR of the noisy seismic volume, which is set to vary between 5 dB and 10 dB. During the training process, the mean squared error (MSE) between the original and reconstructed seismic volumes will be calculated as the training loss, and the multi-scale structural similarity (MS-SSIM) will be used as metrics to evaluate the network's generalization performance on the test set.

Figure 16 shows the evolution of training loss and test set metrics over the training epochs. The training loss decreases rapidly in the first 25 epochs, and reaches full convergence after 200 epochs. Meanwhile, the reconstruction of seismic volumes in the test set achieves an average MS-SSIM of 0.96, in spite of some minor fluctuations. The reconstruction of a synthetic seismic volume from the test set is demonstrated in Figure 17a. Seismic events, including those related to the paleochannels (indicated by the yellow arrows) are mostly reconstructed. However, as shown in the residual volume, random noise and some weak seismic reflections within geologic layers (i.e., between seismic events) are not fully recovered. The reconstruction results of the three field seismic volumes with meandering channels, tributary channel network and submarine canyon are respectively demonstrated in Figure 17b, c, and d. The general patterns (i.e., geometries, relative seismic amplitudes) of the seismic events and paleochannels have been successfully reconstructed. However, we can see from the residual volumes that many detailed seismic reflections related to the geologic layers and paleochannels have not been recovered, especially for the seismic volumes from the Parihaka survey (Figure 17b and c). Table 1 lists the metrics of the autoencoder for reconstructing the synthetic and field seismic volumes shown in Figure 17. The reconstruction of the Parihaka seismic volumes (Figure 17b and d) is less accurate compared to that of the synthetic seismic volume (Figure 17a). However, the autoencoder is capable of reconstructing the NW seismic volume (Figure 17c) with a quality comparable to that of the synthetic seismic volume.

**Table 1.** Metrics of the autoencoder for reconstructing synthetic and field seismic volumes.

| Seismic/channel type | Source | MS-SSIM* ↑ | MSE* ↓ |
|---|---|---|---|
| Synthetic/assorted (Figure 17a) | *cigChannel* Dataset | 0.93 | 0.17 |
| Field/meandering (Figure 17b) | Parihaka survey | 0.86 | 0.23 |
| Field/tributary (Figure 17c) | NW survey | 0.95 | 0.04 |
| Field/submarine (Figure 17d) | Parihaka survey | 0.79 | 0.23 |

[*] MS-SSIM: Multi-scale structural similarity

[*] MSE: Mean squared error

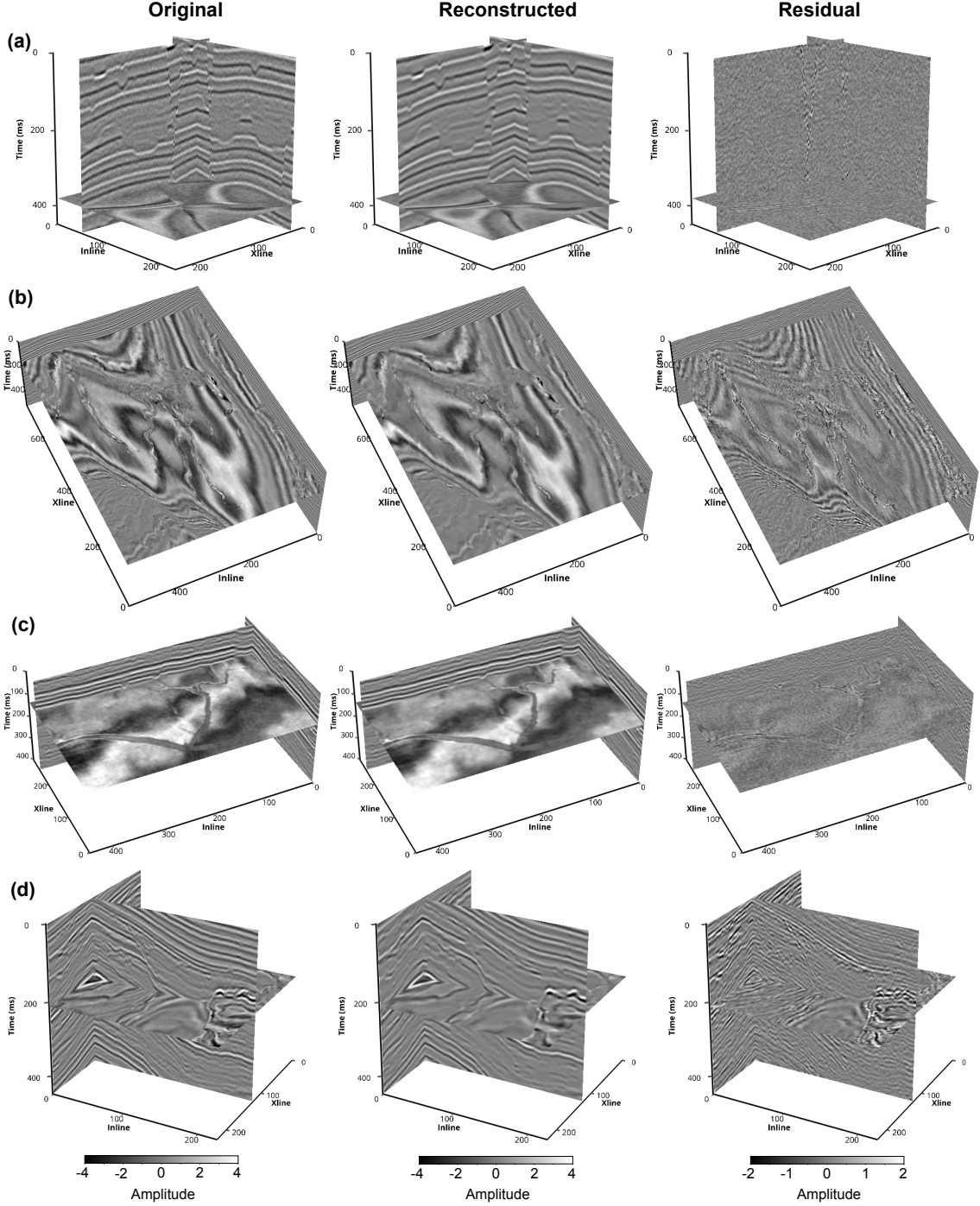

**Figure 17.** The original, reconstructed and residual volumes of synthetic and field seismic data. (a) Synthetic seismic volume with assorted channels. Field seismic volumes with (b) meandering channels, (c) a tributary channel network and (d) a submarine canyon.

The difference in reconstruction performance on field seismic volumes is likely related to the variability in seismic data. Compared with the two Parihaka seismic volumes (Figure 17b and d), the NW seismic volume (Figure 17c) has less variations in seismic amplitude along seismic events, and the seismic amplitude within paleochannels is relatively uniform. These characteristics are similar to the synthetic seismic volumes, and therefore the autoencoder can reconstruct the NW seismic volume as effectively as the synthetic ones. In conclusion, the synthetic seismic volumes have captured the general patterns in field seismic data, such as the geometries of structures and paleochannels. However, they cannot capture the detailed variations in seismic data that are related to wave propagation and changes in rock properties. This may lead to generalization issues for deep learning models trained on this dataset when applied to field seismic volumes with significant variability. Applying more realistic seismic forward modeling methods such as full-waveform modeling and considering the variations in rock properties within geologic layers and paleochannel facies could help improve the plausibility of the synthetic seismic volumes.

## 5.2 Limitations of the dataset

Although the application of the *cigChannel* dataset has shown the capability of training deep learning models to identify paleochannels in field seismic volumes, there are several limitations of this dataset that users should be aware of. The first one lies in the diversity of terrestrial channel and submarine canyon models. The widths of terrestrial channels are set to be relatively small ($\leq 500$ m) in order to be more distinguishable with submarine canyons. However, much wider terrestrial channel systems (e.g., $\geq 1$ km) have also been reported (Gibling, 2006), which could be comparable in size with a relatively narrow submarine canyon such as the La Jolla canyon (Paull et al., 2013). Therefore, if the aim is to train a deep learning model to differentiate between terrestrial channels and submarine canyons, then the model trained on the assorted channel subset may face challenges when distinguishing small submarine canyons from large terrestrial channels. Moreover, as we mentioned, our modeling of submarine canyons aims to replicate the characteristics of the submarine channel-levee system, which requires enough fine-grained sediments to form levees. Relatively coarse-grained sediments (e.g., conglomeratic channel lag deposits) that correspond to a sandier depositional environment are not captured in our submarine canyon models. Consequently, deep learning models trained on the submarine canyon subset may struggle to accurately identify submarine canyons that contain a significant amount of coarse-grained sediments.

The second limitation concerns the realism of seismic impedance within channels. We assign a relatively uniform seismic impedance to terrestrial channels, introducing slight random perturbations to capture natural variability. The seismic impedance of these channels is determined based on a predefined contrast with the overlying layers. However, these simplifications reduce the realism of the impedance representation. In reality, terrestrial channel fills exhibit variations in facies and lithologies (Miall, 2014; Mueller and Pitlick, 2013), which can result in considerable seismic impedance heterogeneity. Although under certain circumstances this heterogeneity could be diminished due to the relatively small size of terrestrial channels and the inherent limitations of seismic resolution, assigning a relatively uniform impedance to terrestrial channels limits the comprehensiveness of their seismic response. As a result, deep learning models trained on the subset of meandering channels or tributary channel networks may face challenges to accurately identify channels that exhibit heterogeneous seismic amplitudes, such as the example shown in Figure 12. Additionally, for submarine canyons, seismic impedance variations related to grain

size distribution within sedimentary facies are not accounted for. The spatial transition from coarse-grained sediments in the channel thalweg to fine-grained sediments along the channel margins (Jobe et al., 2017) is not represented in our impedance models, which further limits the diversity and realism of the synthetic seismic volumes. Consequently, deep learning models trained on the submarine canyon subset may face generalization challenges when applied to identify submarine canyons in field seismic volumes.

The third limitation relates to the realism of non-channel areas in the synthetic seismic volumes. In addition to not fully capturing various characteristics of wave propagation due to the use of 1D convolution for seismic synthesis, the synthetic seismic volumes also lack structural diversity and stratigraphic variability. While folds and faults are included, their scales are enlarged to be comparable to the horizontal extent of the seismic volumes (i.e., 6.4 km). Small-scale (e.g., hundreds of meters) structural deformations, particularly those forming localized U- or V-shaped geometries, are not incorporated,

despite their common occurrence in field seismic volumes. Consequently, deep learning models trained on our dataset may struggle to distinguish between small-scale concave structures and U- or V-shaped channels, which could lead to false positive results. Moreover, each layer in the seismic impedance model is assigned a uniform thickness and a relatively consistent seismic impedance, resulting in a lack of stratigraphic variability in the synthetic seismic volumes. Given this limitation, it is not surprising that a deep learning model trained on our dataset may infer that the primary distinction between channel

and non-channel areas is the presence of stratigraphic variability. This inference arises because, in the synthetic seismic volumes, channels—particularly submarine canyons—are the only structures exhibiting such variability. However, in field seismic volumes, stratigraphic variability is widespread among non-channel areas. Consequently, deep learning models trained on our dataset may produce false positives in non-channel areas with significant stratigraphic variability.

## 6  Conclusions

In this paper, we present a workflow for generating a large number of 3D synthetic seismic volumes containing paleochannels along with their corresponding segmentation labels. Using this approach, we construct the *cigChannel* dataset, which comprises 1,600 seismic volumes featuring three distinct types of paleochannels. This dataset is designed to address the scarcity of training data for deep learning-based paleochannel identification in seismic volumes. Compared to previously used datasets (Pham et al. (2019) and Gao et al. (2021)), the *cigChannel* dataset offers a more diverse and comprehensive collection of paleochannels.

The effectiveness of this dataset is demonstrated through its application to three field seismic volumes, where three simplified U-Nets, trained on the *cigChannel* dataset, successfully identifies paleochannels with promising results. This highlights the feasibility of using synthetic data to train deep learning models for paleochannel identifications, bridging the gap between limited field seismic volume annotations and the need for efficient and robust seismic paleochannel interpretation. Beyond providing a rich source of training samples for deep learning models, the *cigChannel* dataset and its generation workflow

hold potential for advancing seismic modeling techniques and supporting educational applications. For example, rock physics models incorporating fluvial or turbiditic facies could be developed to evaluate new seismic modeling approaches, while the synthetic seismic volumes could serve as effective tools for demonstrating the influence of geological heterogeneities on

seismic data. However, synthetic seismic volumes in the *cigChannel* dataset still lack the diversity and realism of field seismic volumes, primarily due to the simplifications of channel modeling, seismic impedance representation, and the synthesis of seismic volumes.

In the future, we aim to enhance our workflow to improve the realism and diversity of the generated seismic volumes. Terrestrial channels will be modeled using stratigraphic approaches to better capture sedimentary processes, thereby enhancing geological realism. The dataset will also be expanded to include a broader range of channel types, such as braided channels and deltaic systems, further increasing its diversity. To improve seismic impedance modeling, we plan to account for grain size distribution and its impact on impedance variations within channel sedimentary facies. Additionally, the current simplistic 1D convolution will be replaced with 3D convolution or full-waveform modeling to better capture the variability in seismic data. These advancements will enhance the geological realism and diversity of our dataset, ultimately improving its effectiveness for deep learning-based seismic paleochannel interpretation.

## 7  Code and data availability

The *cigChannel* dataset (Wang et al., 2024) can be accessed via Zenodo. It has been organized into four subsets, whose links are provided as followed:

1. Meandering channels: https://doi.org/10.5281/zenodo.11078794;

2. Tributary channel networks: https://doi.org/10.5281/zenodo.11073030;

3. Submarine canyons: https://doi.org/10.5281/zenodo.11079950;

4. Assorted channels: https://doi.org/10.5281/zenodo.11044512.

The codes for dataset generation and the U-Net model used for paleochannel identification are available on GitHub at https://github.com/wanggy-1/cigChannel.

The three seismic volumes presented in the Application section are available for download from the following links:

1. Meandering channel example: https://drive.google.com/file/d/1ItOmdluWUfApzamA4mCeJNhnz_CYUZuf/view?usp= drive_link;

2. Tributary channel network example: https://drive.google.com/file/d/1l4-gBRE-SEoQkx7souERjtiRLpyABrJ-/view?usp= drive_link;

3. Submarine canyon example: https://drive.google.com/file/d/1qxO8-onWFlffp7t3UHMtm-rHUmkkMvQx/view?usp=drive_ link.

 **Appendix A: Channel modeling parameters**

**Table A1.** Channel modeling parameters.

| Channel type | Parameter | Value | Reference |
|---|---|---|---|
| Meandering channel | Width | 200 m - 500 m | 30 m - 15 km (Gibling, 2006) |
| | Maximum depth | 20 m - 50 m | 1 m - 38 m (Gibling, 2006) |
| | Strike | N0°E - N360°E | |
| | Migration rate constant | 40 m/yr - 50 m/yr | Exaggerated to accelerate simulation; reference range: 0.5 m/yr - 15 m/yr (Donovan et al., 2021; Schook et al., 2017; Heo et al., 2009) |
| | Dimensionless Chezy's friction factor | 0.06 - 0.08 | Exaggerated to accelerate simulation; reference range: 0.002 - 0.005 (Chow, 1988) |
| | Iteration time step | 0.1 yr | |
| | Number of iterations | 1000 - 2000 | |
| Tributary channel network | Maximum width | 200 m - 400 m | 10 m - 1000 m (Trigg et al., 2012) |
| | Width/depth ratio | 10 - 12 | 2 - 870 (Gibling, 2006) |
| | Maximum number of iterations | 8192 | |
| | Number of particles for early-termination | 0 | |
| Submarine canyon | Channel width | 300 m - 400 m | 195 m - 6.8 km (Shumaker et al., 2018) |
| | Maximum depth | 30 m - 40 m | 4 m - 132 m (Shumaker et al., 2018) |
| | Strike | N0°E - N360°E | |
| | Migration rate constant | 50 m/yr - 60 m/yr | Exaggerated to accelerate simulation; reference range: 2 m/yr - 14 m/yr (Biscara et al., 2013) |
| | Dimensionless Chezy's friction factor | 0.07 - 0.08 | Exaggerated to accelerate simulation process; reference range: 0.002 - 0.005 (Chow, 1988) |
| | Iteration time step | 0.1 yr | |
| | Number of iterations | 500 - 2000 | |
| | Natural levee deposition rate | 5 m/yr | Exaggerated to accelerate simulation; reference value: 0.66 m/kyr (Allen et al., 2022) |
| | Natural levee width | 6 km - 8 km | Restrained to fit model's extension; reference range: 25 km - 40 km (Klaucke et al., 1998) |
| | Channel incision rate | 8 m/yr | Exaggerated to accelerate simulation; reference value: 80 m/myr (Englert et al., 2020) |
| | Channel aggradation rate | 8 m/yr | Exaggerated to accelerate simulation; reference value: 300 m/myr (Englert et al., 2020) |

# Appendix B: Components of the *cigChannel* dataset

**Table B1.** Components of the *cigChannel* dataset.

| Name | Sample amount | Contents | Features | Example |
|------|---------------|----------|----------|---------|
| Meandering channel subset | 400 | 1. Seismic volumes<br>2. Binary label volumes<br>3. Seismic impedance volumes | 1. Meandering channels<br>2. Horizontal, inclined, folded and faulted structures<br>3. Noise-free |  |
| Tributary channel network subset | 400 | 1. Seismic volumes<br>2. Binary label volumes<br>3. Seismic impedance volumes | 1. Tributary channel networks<br>2. Horizontal, inclined, folded and faulted structures<br>3. Noise-free |  |
| Submarine canyon subset | 400 | 1. Seismic volumes<br>2. Binary label volumes<br>3. Seismic impedance volumes<br>4. Sedimentary facies volumes | 1. Submarine canyons<br>2. Horizontal, inclined, folded and faulted structures<br>3. Noise-free |  |
| Assorted channel subset | 400 | 1. Seismic volumes<br>2. Multi-class label volumes<br>3. Seismic impedance volumes | 1. Meandering channels, tributary channel networks and submarine canyons<br>2. Horizontal, inclined, folded and faulted structures<br>3. Noise-free |  |

## Appendix C: Demonstration codes of the dataset generation workflow

```
1: # Import all functions.
2: from functions import *
3:
4: # Number of models.
5: n_model = 400
6: # Data generation.
7: for i in range(n_model):
8:     # Initialize the model.
9:     model = GeoModel()
10:    # Assign P-wave velocities.
11:    model.add_vp()
12:    # Add meandering channels.
13:    model.add_meandering_channel()
14:    # Add tributary channels.
15:    model.add_tributary_channel()
16:    # Add submarine canyons.
17:    model.add_submarine_canyon()
18:    # Add inclination.
19:    model.add_dipping()
20:    # Add folds.
21:    model.add_fold()
22:    # Add faults.
23:    model.add_faults()
24:    # Resampling model's z-coordinates.
25:    model.resample_z()
26:    # Compute P-wave impedance.
27:    model.compute_Ip()
28:    # Compute reflection coefficients.
29:    model.compute_rc()
30:    # Make synthetic seismic data.
31:    model.make_synseis()
32:    # Save data.
33:    model.Ip.tofile()  # Impedance volume.
34:    model.seismic.tofile()  # Seismic volume.
35:    model.seis_label.tofile()  # Channel label volume.
36:    model.facies.tofile()  # Sedimentary facies volume.
```

# Appendix D:  Parameters of the seismic impedance model and Ricker wavelet

**Table D1.** Parameters of the seismic impedance model, Ricker wavelet and their reference values.

| | Parameter | Value |
|---|---|---|
| Model extension | X | 0 m - 6400 m |
| | Y | 0 m - 6400 m |
| | Z | 0 m - 1280 m |
| | Grid spacing | 25 m × 25 m × 5 m (X × Y × Z) |
| Layer | Seismic impedance | 7000 m/s.g/cm$^3$ - 16000 m/s.g/cm$^3$ |
| | Impedance perturbation | 300 m/s.g/cm$^3$ - 500 m/s.g/cm$^3$ |
| | Thickness | 60 m - 150 m |
| Meandering channel | Impedance contrast with the overlying layer ($\varepsilon$) | 0 - 1 |
| Tributary channel | Impedance contrast with the overlying layer ($\varepsilon$) | 0 - 1 |
| Submarine canyon | Point bar impedance | 6000 m/s.g/cm$^3$ - 8400 m/s.g/cm$^3$ |
| | Natural levee impedance | 8400 m/s.g/cm$^3$ - 14400 m/s.g/cm$^3$ |
| | Abandoned meander impedance | 8400 m/s.g/cm$^3$ - 14400 m/s.g/cm$^3$ |
| Ricker wavelet | Peak wavenumber | 20 km$^{-1}$ - 60 km$^{-1}$ |

*Author contributions.* Guangyu Wang developed the Python package for the dataset generation workflow and wrote the manuscript. Xinming Wu initiated the idea of constructing a large-scale paleochannel dataset for deep learning–based seismic interpretation and co-wrote the manuscript. Wen Zhang conducted the training and application of U-Net models for paleochannel identification in field seismic volumes.

*Competing interests.* The authors declare no competing interests.

*Acknowledgements.* We acknowledge the USTC supercomputing center for providing computational resources for this project and Jintao Li for providing the Python package *CIGVis* (Li et al., 2025) to visualize the 3D seismic volumes. We appreciate the valuable feedback and suggestions from Andrea Rovida, Samuel Bignardi and one anonymous referee, which have greatly enhanced the clarity and rigor of this work. We would also like to thank Hang Gao and Jiarun Yang for their useful suggestions on the training strategy of the U-Net.

*Financial support.* This work was financially supported by the National Science Foundation of China under Grant 42374127.

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
