# Peer review of "cigChannel: A large-scale 3D seismic dataset with labeled paleochannels for advancing deep learning in seismic interpretation"

_Earth System Science Data, 2024_

## Author Response (AR1)

**Response Letter**

**Responses to comments from reviewer 1**

Dear Anonymous reviewer,

Thank you for taking the time to provide such detailed and constructive feedback on our manuscript. It is really helpful to improve the quality and clarity of our work. In response, we have removed the claim that our dataset is realistic and comprehensive and have incorporated faults into the dataset to enhance its usefulness. Additionally, we have corrected the terminology from "distributary channels" to "tributary channels". Furthermore, we have explicitly acknowledged the limitations of our dataset and discussed their potential impact. To strengthen the plausibility of our model, we have cited additional geological literature. Moreover, we have conducted a data reconstruction analysis to validate the reliability of our dataset and have further highlighted the broader value of our workflow beyond deep learning development.

**Responses to major comments:**

We admit that the synthetic seismic volumes in this dataset are still not as realistic and comprehensive as field seismic volumes. We have removed the claim of being realistic and comprehensive. The limitations of our dataset are now clearly acknowledged in the Discussion section, as is its impact on deep learning models when applied to real data.

The reason of using Soillib to instead of more established models to simulate tributary channels (Using the term of "distributary channels" was a mistake. We have corrected it in our revised manuscript) is because it is relatively easy to control the channel's width and depth. Established models such as Landlab, Fastscape, and Badland can simulate realistic tributary river systems. However, we found them difficult to control the channel's geometry. For the purpose of providing more variability to the channel's geometry, we ultimately decide to use the Soillib package. Nevertheless, we acknowledge that our workflow can only simulate morphologically reasonable channels, which are not as geologically realistic as those generated by the stratigraphic models. We have also acknowledged that our workflow does not capture a large part of variability in field seismic data, and the impact is discussed in the revised manuscript. Additionally, we have now added faults to our dataset, which would hopefully improve the usefulness of this dataset.

More geological literatures have been cited to support the plausibility of our models. We admit that it was a mistake to claim that we use Soillib to simulate distributary channels. It should be tributary channels and we have corrected it in our revised manuscript. The limitations of the workflow and dataset have now been clearly acknowledged and discussed in the revised manuscript, including in the abstract.

As for whether this dataset is plausible enough to be used confidently for the validation of new methods. We have conducted the data reconstruction experiment as you suggested. An autoencoder was trained on our synthetic seismic volumes and applied to reconstructed the three field seismic volumes demonstrated in the Application section. The results show that the reconstruction performance of the autoencoder depends on the variability of field seismic volume. For a field seismic volume with relatively simple seismic patterns (e.g., less variations along seismic events), such as the demonstrated volume with tributary channels, the autoencoder can reconstruct it with a quality comparable to that of a synthetic seismic volume. However, for field seismic volumes with significant variability, such as the two Parihaka volumes, the reconstruction accuracy is lower than that of the synthetic volumes. We conclude that our synthetic volumes have captured the general patterns in field seismic volumes (e.g., geometries of structures and paleochannels), but not the detailed variations in seismic data that are related to wave propagation and changes in rock properties. Therefore, methods validate on our dataset may face challenges when applied to field seismic volumes with significant variability. This limitation has now been acknowledged in the revised manuscript.

Lastly, we have also reflected the value of our workflow and dataset beyond the development of deep learning algorithms. For instance, rock physics models incorporating fluvial or turbiditic facies could be developed to evaluate new seismic modeling approaches, while the synthetic seismic volumes could serve as effective tools for demonstrating the influence of geological heterogeneities on seismic data. To promote the adoption and further development of our workflow, we are transforming it into a well-documented Python package, which will soon be available on PyPi.

**Responses to specific comments:**

**Abstract**

Line 8: "to most researchers" instead of "to most of the researchers".

We are sorry for this mistake. It has been corrected.

Line 9: If it's geologically reasonable then it's not realistic. Something realistic represents a system accurately, while reasonable means that it's a good enough approximation. In this case we're in the later: the geology here isn't realistic at all, but considering the lack of resolution of seismic data it's a good enough approximation to get some valuable insights from deep learning.

We agree. We now recognize that this dataset is still far from being geologically realistic, so we have removed that claim.

Line 13: What does it mean to "perform well"? Better be specific and quantitative there.

Thanks for your advice. We have added the F1 scores between U-Net and human-made channel interpretation.

Line 14: How many seismic volumes?

We have added the number of seismic volumes.

Line 14: "which indicates the diversity and representativeness of the dataset" Nothing in the abstract suggests that you can conclude that.

You're right, this claim is too assertive. It has been removed.

Line 15: That's great!

**1. Introduction**

Line 20: Considering that we're already experiencing the consequences of climate change, that focus solely on hydrocarbons is unfortunate. Paleochannels are good reservoirs, so they are also valuable in hydrogeology, hydrothermal production, and mining.

We have added these aspects (see lines 17-20 in the revised manuscript).

Line 27: "have been developed" instead of "are developed".

We are sorry for this mistake. It has been corrected.

Line 35: More and more seismic data are being released by government agencies (in Australia, the Netherlands, New Zealand, ...), so lack of access is not as true as it used to be. A key issue remains processing: those data can be raw or not completely processed (e.g., not depth converted), so it's difficult for non-specialists to reuse them. And then, as the authors rightfully mentioned, there's the difficulty in interpreting the data.

Yes, you are right. A sufficient number of open-access field seismic volumes have been released by various organizations. We have revised this claim (see lines 35-36 in the revised manuscript).

Line 39: I wouldn't say that it's not an option, it's just an expensive one, prone to uncertainties (in the processing for instance) and to biases (see for instance https://doi.org/10.1130/GSAT01711A.1).

We agree. We have corrected it.

Line 51: "massive-scale" is exaggerated. It's a relatively large dataset for the subsurface, but it's nothing compared to datasets from the deep learning community. And even in the subsurface much larger datasets have been released before (see https://doi.org/10.5194/essd-14-381-2022 for instance).

Yes, it is relatively small compared to the Noddyverse dataset. We have changed "massive-scale" to "large-scale", including in the title.

Line 65: Maybe mention the link to the GitHub repository also here?

The link has been added.

**2. Dataset generation workflow**

Line 67: This sentence is a bit convoluted, with several repetitions that can be avoided (i.e., "generation" then "generating", "elaborate" then "explain details").

We have revised this sentence to avoid repetitions (see lines 67-70 in the revised manuscript).

Line 72: Are meandering channels the most common river channels? I'm not convinced of that, it would be better to support this claim with a reference.

We acknowledge that this claim is problematic. It has been revised to "Meandering channels are among the commonly observed river channels in seismic volumes", followed by several supporting references (see lines 72-73 in the revised manuscript).

Line 86: Any reference to support those two shapes? It would support the claim of realism much better to show that this is indeed what we observe in nature.

Yes. Thanks for pointing this out. Supporting references have been added (see lines 89-90 in the revised manuscript).

Line 105: I couldn't find a paper describing how this model works in detail. The key problem here is that the channels shown in figure 3 look nothing like deltaic channels, but more like a continental river system (and those look more like tributaries, not distributaries). Many models have been developed to simulate such systems (see Landlab, Fastscape, Badlands) based on laws that only approximate the physics of overland flow, erosion, and deposition but have been validated to some degree and can be fast depending on the processes included and the implementation. So why not use those models? Regarding deltas, DeltaRCM developed by Liang et al. (2015) is a valid candidate, even if it's still a bit slow. I'm not sure if Sedflux could be an option (https://doi.org/10.1016/j.cageo.2008.02.013), but it shows that deltas are more than just channels, and that will impact the seismic data.

We are sorry for this mistake. It is true that the Soillib package simulates terrestrial tributary river networks. We have changed all content related to distributary channels to tributary channels. There is no published paper about the modeling approach related to the Soillib package. Details of the modeling approach are described in an online blog (https://nickmcd.me/2020/04/10/simple-particle-based-hydraulic-erosion). We appreciate your recommendation of those landscape evolution models like Landlab, Fastscape and Badlands. We have investigated into these models. They indeed can simulate very realistic tributary river systems and can be fast. However, it's difficult to control the width and depth of the generated river channel. Using Soillib is able to control the channel's width and depth, therefore offers more varieties to the channel's geometry. Nonetheless, using Soillib can only generate morphologically reasonable tributary channels. They lack geologically

realism compared to those generated by stratigraphic models. This point has now been acknowledged in the revised manuscript (see lines 132-136 in the revised manuscript).

Line 127: Not just any sediment, you need enough fine sediments to build levees, so meandering turbiditic channels correspond to a quite specific depositional environment. This could restrict the comprehensiveness of the dataset, which won't capture sandier environments.

Yes, we agree. We have added a statement that in this work we aim at modeling meandering submarine channels that carry enough fine sediments to build levees (see lines 140-142 in the revised manuscript). We acknowledge that this could limit the comprehensiveness of the modeled submarine channel, which is reflected in the newly added Discussion section (see lines 343-355 in the revised manuscript).

Line 130: The main difference is the scale: submarine channels are much wider, deeper, and longer than their terrestrial counterparts, which isn't really clear from figure 5. The lack of spatial scale also doesn't help in assessing the plausibility of the 3D structures, especially relative to one another.

Thanks for pointing this out. The spatial scale has been added to all figures.

Line 152: I would say abandoned meander instead of oxbow lake, which fits more a continental setting.

We have corrected it from "oxbow lake" to "abandoned meanders".

Line 164: Channels in general have different facies (point bars, levees, crevasse splays, abandoned meanders, abandoned channels). You take that into account when selecting the impedance for submarine channels but not the others, why is that? And what's the impact of that choice? On top of that, variations in grain size distribution within a facies leads to variations of impedance, so why using a uniform impedance, which isn't realistic? And this is excluding the effect of burial and diagenesis.

It is true that both terrestrial and submarine channels have different facies. The reason why we only take facies into account when selecting impedance for submarine channels (Actually we mean submarine canyons carved out by the movement of meandering submarine channels. We have corrected it in the revised manuscript) is that under seismic resolution, facies are more prominent in submarine canyons due to their large sizes. We acknowledge that even for terrestrial channels with relatively small scales, variations in facies and grain size within the channel will lead to variations in impedance. However, for terrestrial channels, we aim to detect their last channel form. Point bars are more relevant to the channel migration process rather than to the last channel form, so are levees and abandoned meanders. On the contrary, these facies generally can be hosted in submarine canyons, so we take them into account when selecting impedance for submarine canyons. Nonetheless, we acknowledge that using a relatively uniform impedance within a facies is not realistic. This choice will certainly impact network's performance on detecting channels with varying impedance. However, it is difficult to consider grain size distribution and effect of burial and diagenesis when modeling channels. We have now discussed this limitation of our dataset in the newly added Discussion section (see lines 356-369 in the revised manuscript).

Line 177: So this is a 1D convolution? I get that this is a common and simple approach, but it's not really realistic either (see for instance https://doi.org/10.1111/1365-2478.12936 or https://library.seg.org/doi/full/10.1190/1.2919584, and https://doi.org/10.1190/geo2021-0824.1 for an application to seismic data interpretation using deep learning).

Yes. It is a 1D convolution. We acknowledge this is not realistic and have now discuss its limitation in the newly added Discussion section (see section 5.1 in the revised manuscript).

Figure 8: What's the spatial scale of the 2D sections? This comment stands for almost all the figures, but here in particular because we can't compare to the wavelet without a clear scale. And how do the peak wavenumbers relate to the usual values in Hz?

We are sorry for this. The spatial scale has been added to all figures. The values of peak wavenumbers are identical to that of peak frequencies (1 km$^{-1}$ = 1 Hz, see https://doi.org/10.1190/geo2020-0745.1). We have added a note for that in the figure caption.

**3. Results**

Line 219: No faults? How does that impact the usefulness of the dataset?

Thanks for pointing this out. We have added faults into our dataset.

Line 228: Actually size and aggradation are the only differences between fluvial and turbiditic channels in your simulations, since the model for meandering is the same. So a deep learning model trained on your dataset might struggle with small turbiditic systems and large fluvial systems.

Yes, it can happen. This limitation is now acknowledged and discussed in the newly added Discussion section (see lines 345-350 in the revised manuscript).

Lines 228-229: That doesn't really explain why they are so much larger. Overall I see very little geological literature cited to support the plausibility of the models, which is unfortunate.

We are sorry for this mistake. We have provided references to support the plausibility of setting submarine canyons wider and deeper than terrestrial channels (see lines 240-242 in the revised manuscript).

Line 235: Any reference to show how to do that? Any reference for the weighted loss function?

We suggest to solve the class imbalance problem by using the class-balanced cross-entropy loss function during network training. A reference for that is also given (see lines 249-251 in the revised manuscript).

**4. Applications**

Line 249: It would be nice to add a link to that dataset.

*Thanks for your suggestion. We have added the link.*

Line 251: This isn't quantitative, which is unfortunate. It would have been much better to compare to a human-made interpretation, especially since here you have nice channels that look easily interpretable, and measure different metrics such as precision and recall. There seems to be a lot of false positives in the deep-learning interpretation. I realize that the manuscript doesn't aim at developing a deep-learning model for channel interpretation, but are the false positives due to a not-so-optimal model or a not-so-optimal dataset? Not having any validation metric for the training of the deep-learning model doesn't help to assess this.

*Thanks for your suggestion. We have manually interpreted the channels in all three field seismic volumes, and calculated the F1 scores between the U-Net and human-made interpretation results (see lines 275, 287, and 293). We have also demonstrated the F1 scores of the test sets during the training of the U-Nets (Figure 11 in the revised manuscript). They reach an average score of 0.9 at the end of training. However, the average F1 scores for channel interpretation in field seismic volumes is 0.63. The presence of false positives is certainly one of the reasons why the average F1 score of field seismic volumes is lower than that of the synthetic ones. Since the U-Net performs well in identifying channels in synthetic seismic volumes but less effectively in field seismic volumes, these false positives are more likely due to a not-so-optimal dataset instead of a suboptimal model.*

Line 256 and 261: Are those seismic volumes open? If not, that part of the manuscript is irreproducible.

*Yes, they are open. We now have provided download links for each of those seismic volumes in the Code and data availability section (line 420).*

Line 270: This is really a strong limitation considering that faults are ubiquitous in the subsurface, can have a big impact on applications, and that there has been a lot of studies similar to yours for faults already, so methods to introduce faults already exist (e.g., https://doi.org/10.1190/geo2021-0824.1).

*Yes, we agree. We have now added faults to our dataset.*

Line 279: That's not quite true: you're proposing a benchmark dataset (lines 5, 59, 290), so a standard to compare (future) methods. How can your dataset become a standard if it excludes a basic configuration of the subsurface (faulted domains)?

*We agree. We have now included faults into our dataset to make it more comprehensive.*

5. Conclusions

Line 284: What are the predecessors actually? You've never mentioned any.

*We are sorry for this mistake. We have added elaboration of those predecessors (see lines 388-389).*

Line 285-286: I'm dubious of the claims of realism and diversity, which aren't well supported by the manuscript. That doesn't mean that this dataset cannot be useful, but I expect more openness on the limitations, which is essential if this is to be used as a benchmark.

We admit that our dataset still lacks realism and diversity of field data. The limitations have now been clearly acknowledged and discussed in the newly added Discussion section.

Table A1: It would be much better to have some justification for those values, either in an extra reference column or in the text. Channels can be smaller than 200 m and larger than 500 m (see https://doi.org/10.2110/jsr.2006.060).

Thanks for your suggestion. We have added an extra reference column to justify those values.

Once again, we sincerely appreciate your time and effort in reviewing our manuscript. Your valuable feedback has been instrumental in refining our work.

Best regards,

Guangyu Wang
Xinming Wu
Wen Zhang

**Responses to comments from reviewer 2:**

Dear Samuel Bignardi,

We deeply appreciate your time and effort in reviewing our manuscript. Your comments have been really helpful in refining our work. In response, we have conducted reliability tests on our dataset and the weak aspects have now been clearly disclosed.

**Responses to major comments:**

We admit that using 1D convolution to construct synthetic seismic volumes is very simplistic, as it ignores many aspects of realistic wave propagation. The impact of such choice has been disclosed by a data reconstruction test (see Figure 15 in our revised manuscript). It shows that an autoencoder trained on our synthetic seismic volume cannot fully reconstruct detailed variations in field seismic data. This means that our synthetic data fails to capture the detailed seismic variations in field data, which could be related to wave propagation. This limitation has now been clearly acknowledged in our revised manuscript (see line 337). Although using realistic seismic forward model (e.g., full-waveform simulation) could capture many characteristics of wave propagation, it would take tremendous amount of computational time to finish the process from wavefield simulation to seismic migration, considering that we are building many 3D seismic models. Nonetheless, we agree with you that this computational time is worth paying. We plan to adopt more realistic seismic forward model to improve the realism of the next version of our dataset.

As for the representation of subsurface models, we agree that using parallel-layered models and place channels at the boundary could introduce an implicit geometrical bias to the CNN, making it believe that any discontinuity on the parallel layer should be channels, and eventually mistakes the fault for a channel. Therefore, we have now added faults to the parallel-layered models (see Figure 8 and 9 in our revised manuscript) and re-trained the CNN on the new dataset. Now the CNN can correctly identify faults as non-channel, as shown in Figure 12 in the revised manuscript.

For the reliability test, we believe a major purpose is to validate if our current seismic simulation (1D convolution) could capture the characteristics of realistic wave propagation, as a full-waveform simulation does. Therefore, we directly validate our simulation against real seismic data by training an autoencoder to reconstruct seismic volumes using our synthetic seismic volumes. If the autoencoder could reconstruct real seismic volumes as well as the synthetic ones, this would mean that our dataset fully captures the characteristics of real wave propagation. Otherwise, it suggests room for improvement. The results (Table 1 and Figure 17 in the revised manuscript) indicate that the autoencoder effectively reconstructs the general textures in field seismic volumes but struggle to capture detailed seismic variations. These detailed seismic variations are likely an effect related to wave propagation, which are missing from our synthetic seismic volumes.

We have also performed a reliability test to assess the network's performance (see line 266 in the revised manuscript). We divided the dataset into training and test set. The network was trained on the training set and evaluated on the test set. The results (see Figure 11 in the revised manuscript) indicate that the network perform well on identifying all three types of channels in the synthetic seismic volumes, reaching a high F1 score of approximately 0.9. This could answer the first question in your three examples. If a new subsurface model were produced using our workflow, about 90% of the channel voxels would be labeled correctly.

For the second question, if the network were tested on a simulated seismic volume produced with a third-party algorithm, the quality of the result would depend on whether this algorithm could simulate detailed seismic variations related to wave propagation. If so, then the network may struggle to label channels correctly, as suggested by the relatively poor reconstruction performance of the network on the Parihaka seismic volumes, which have numerous detailed seismic variations (see Table 1 and Figure 17c).

For the last question, we have addressed it by comparing the channel segmentation results obtained from the network and human interpretation across three field seismic volumes. (see Figure 12, 13, and 14 in the revised manuscript).

**Responses to specific comments:**

Line 2: "The data-driven deep learning methods are …", remove "The"

Thank you for pointing out this mistake. It has been removed.

Line 6/7: "Manually labeling 3D channels in field seismic volumes can be a tedious and subjective work …"
In fairness, there is a large subjectivity with CNNs as well, although it comes in a different form. For example, what CNN architecture should we use? How many layers should we employ? What activation function, and what learning rate to adopt? These are ALL subjective choices.

We agree. The training phase of deep learning models has many subjective choices which can affect the model's prediction result. The point we try to convey here is that the subjectivity of human interpreter could lead to wrong interpretation of paleochannels. If we use these wrong interpretations as labels to train a deep learning model, we cannot expect it to have a satisfying performance. (see lines 6-7 in the revised manuscript)

Line 27: "To address those issues, automatic paleochannel identification methods based on 3D convolutional neural networks (CNNs) (Pham et al., 2019; Gao et al., 2021) are developed."
"To address those issues, automatic paleochannel identification methods based on 3D convolutional neural networks (CNNs) (Pham et al., 2019; Gao et al., 2021) HAVE BEEN developed."

We are sorry for this mistake. It has been corrected.

Line 28: "They treat paleochannels as bodies rather than slices as human interpreters typically see,"
I disagree. Human interpreters may visualize slice-wise information but they bear the 3D model in mind. Plus, (auspicably) they regard the data with expertise matured in years of training in geology.
I would change the sentence, to something like: "They have the advantage of handling paleochannels according to their 3D nature, as opposed to the slice-by-slice visual investigation of a human operator."

We agree. Even though a paleochannel is visualized slice-wise, an experienced interpreter gets more information beyond those slices. We appreciate your suggestion. This sentence has been changed (see lines 28-29 in the revised manuscript).

Line 33: "..., currently there is no publicly available dataset of field seismic…"
Remove "currently".

It has been removed.

Line 41/42: "… allowing us to tailor the objectives that we want the network to learn"
Better: "… allowing us to tailor the features that our network will learn to segment"

Many thanks for your suggestion. It has been changed.

Line 53/54: "the modeling methods developed by Howard and Knutson (1984), McDonald (2020) and Sylvester et al. (2011), respectively"
A summary of this modeling technique would be welcome to people unfamiliar with those specific articles.

Thank you for this feedback. This sentence has been removed. Details of these modeling technique are elaborated in their respective sections.

Line 75: "We use the open-source Python package meanderpy (Sylvester, 2021) ..."
Is there any limitation in the Sylvester {2021) modeling that could impact the reliability of this dataset? If so, it should be mentioned.

Yes, there is. Meanderpy employs a simple kinematic model that focuses on the influence of river's upstream curvatures on river's migration. It cannot capture complex processes such as compound meander formation without cutoffs. This limitation has been mentioned in the text (see lines 88-89 in the revised manuscript). Thanks for your suggestion.

Line 180 (equation 8). Earlier in the manuscript, when I read "convolution" I implicitly assumed it was performed in the time domain to obtain a seismic volume with the two-way delay time

along the vertical axis. Seeing equation (8) expressed with wavenumber makes me realize that the vertical axis is depth. The fact that volume figures have no axes also contributed to this misunderstanding. It would be advisable to add an explanation of this aspect soon in the main text, perhaps at the point where the first model is described.

We are sorry for causing this confusion. The axes of models and seismic volumes have now been added.

Line 224: "seismic volumes with multi-class channel labels."
What do you mean by "multi-class labels"?
I was under the impression that this paper used just two classes (i.e. "paleochannel" and "background"). Did I miss something?

Yes, this paper focus on distinguishing paleochannels and the background. However, the assorted channel subset also provides three classes of labels: background, terrestrial channels (meandering and tributary channels) and submarine canyons.

Line 231: "Regarding the potential problems of the class imbalance problem and the size discrepancy between terrestrial and submarine ..."
Better something like: "Regarding possible class imbalance problems connected to the size discrepancy between terrestrial and submarine … "

Thank you for your suggestion. It has been changed.

Line 242/243: "Gaussian random noise is added to the seismic volume to make the training process more robust and reduce the tendency towards overfitting."
A better description of this aspect is required.
How was the noise designed?
How were the mean and variance chosen, and why?
Is the noise somewhat spatially coherent, or was it applied only node-wise?
What is the resulting signal-to-noise ratio?

Thanks for pointing this out. A better description of the noise has been added to the manuscript (see lines 259-261 in the revised manuscript): The noise is zero-mean with a standard deviation determined by the expected SNR of the seismic volume. We choose zero-mean noise so that the distribution of seismic data will not be changed after the noise is added. We set the SNR vary between 5 dB and 10 dB, which is a reasonable range for field seismic volumes (see https://doi.org/10.1016/j.jappgeo.2021.104446 and https://doi.org/10.1007/s11770-017-0607-z for examples). The noise is added to the seismic volume node-wise and is not spatially coherent.

Line 276: "… heterogenous seismic amplitude, which is an exceptional case for our dataset."
Better something like: "… heterogeneous seismic amplitude, a feature that could not be learned as it was not present in our training dataset."

Thanks for your suggestion. It has been changed.

Line 277: "Therefore, the identification performance of channels with heterogeneous seismic amplitude would be improved if meandering and distributary channels with heterogeneous seismic amplitude can be included in this dataset."
Better something like: "The identification of such channels would likely be improved were we including such acoustic impedance heterogeneity in the modeling of channels."

It is indeed better than the original sentence. We have changed it according to your suggestion.

Line 278/279: "As we mentioned, these are preliminary tests mainly to find out whether this dataset can help the network discriminate channels and non-channel areas."
The expression "to find out" is suitable for everyday conversation, but not much in a formal text. It would be better something like: "As we mentioned, the rationale to these preliminary tests is mainly to judge whether this dataset can help the network discriminate channels and non-channel areas."

We apologize for this improper expression. It has been corrected.

Once again, we are truly grateful for your thorough and insightful feedback on our manuscript. Your comments have greatly contributed to improving its clarity and overall quality.

Best regards,

Guangyu Wang
Xinming Wu
Wen Zhang

**List of relevant changes:**

1. Title:
   - "Massive-scale" is changed to "Large-scale".

2. Abstract:
   - Revised for better clarity.
   - Limitations of the dataset is acknowledged.

3. Introduction:
   - Revised content according to the specific comments from reviewers.

4. Dataset generation workflow:
   - Corrected the term "Distributary channel" to "Tributary channel".
   - Changed the term "Submarine channel modeling" to "Submarine canyon modeling".
   - Added more references to support the plausibility of models.
   - Declared that the submarine canyon modeling aims to simulate a specific type of submarine canyon related to the submarine channel-levee system.
   - Acknowledged the limitations of the workflow.
   - Revised content according to the specific comments from reviewers.

5. Seismic volume simulation:
   - Added scales to all figures.
   - Added faults to the seismic impedance models.
   - Explained the relationship between peak wavenumber and peak frequency in the caption of Figure 8.

6. Results (Dataset description):
   - Added 100 samples with faults to each subset.
   - Figure 9 is modified to show samples with faults.

7. Applications:
   - Validated the performance of the U-Net by splitting the dataset into training and test set.
   - Added Figure 11 to show the performance of U-Net during training.
   - Elaborated the type of noise added to the samples before training and the signal-to-noise ratio of the noisy training samples.
   - Added comparison between U-Net and human-made interpretation of paleochannels in all three field seismic volumes.
   - Quantified the U-Net performance on field seismic volumes by calculating the F1 score between U-Net and human-made interpretation results of paleochannels.

- Modified Figure 12, 13, and 14 to show the human-made paleochannel interpretation results.

8. Discussion (new section):
   - Designed a data reconstruction experiment to validate the reliability of our dataset.
   - Clearly acknowledged the limitations of the dataset and discussed the possible influence and failure when applying our dataset.

9. Conclusion:
   - Reflected potential application of our dataset beyond AI algorithm development.
   - Clearly acknowledged the limitations of our dataset and outlined our next step to address those limitations.

---

## Author Response (AR2)

**Response Letter**

**Response to comments from reviewer 1**

Dear anonymous reviewer,

Thank you very much for your positive feedback. We greatly appreciate your recognition of our efforts in addressing the comments and improving the manuscript. We fully agree that there will always be room for refining the methodology and improving the quality of the dataset. Your constructive feedback and recognition have been highly encouraging and helpful in guiding us toward our ultimate goal of constructing a realistic and comprehensive dataset to facilitate deep learning-based paleochannel interpretation.

Additionally, we have corrected the reference for Sedsim and revised "its capability" to "the capability" as you suggested.

Once again, we sincerely appreciate your thoughtful review and constructive feedback. Your comments have provided us with valuable insights and guidance for improving our manuscript.

Best regards,
Guangyu Wang
Xinming Wu
Wen Zhang

**Response to comments from reviewer 2**

Dear Samuel Bignardi,

We sincerely appreciate your positive recognition of our revised manuscript. Your feedback on our original submission has been extremely helpful in improving its clarity, fairness, and overall quality. Moving forward, we will continue to explore efficient seismic forward modeling methods to better capture the realistic characteristics of wave propagation.

Additionally, as mentioned in our manuscript (line 232), the subsurface models (seismic impedance and sedimentary facies volumes) have already been published together with the seismic volumes. Nevertheless, to further clarify, we have included supplementary figures to illustrate the subsurface models corresponding to the seismic volumes shown in Figure 9.

Once again, we sincerely appreciate your time and effort in reviewing our manuscript. Your comments have been instrumental in improving its quality.

Best regards,
Guangyu Wang
Xinming Wu
Wen Zhang